# Ultrastructural alterations in the retinal pigment epithelium and photoreceptors of a Stargardt patient and three Stargardt mouse models: indication for the central role of RPE melanin in oxidative stress

Tatjana Taubitz, Alexander V. Tschulakow, Marina Tikhonovich, Barbara Illing, Yuan Fang, Antje Biesemeier, Sylvie Julien-Schraermeyer and Ulrich Schraermeyer

Division of Experimental Vitreoretinal Surgery, Centre for Ophthalmology, University of Tuebingen, Tuebingen, Germany

## ABSTRACT

**Background**. Stargardt disease (SD) is characterized by the accumulation of the age-pigment lipofuscin in the retinal pigment epithelium (RPE) and subsequent neuroretinal degeneration. The disease leads to vision loss early in life. Here, we investigate age-dependent ultrastructural changes in three SD mouse models: albino *Abca4*$^{-/-}$ and pigmented *Abca4*$^{-/-}$ and *Abca4*$^{-/-}$.*Rdh8*$^{-/-}$ mice. Since we found indications for oxidative stress primarily in albino SD mice, we tested RPE melanin for its antioxidative capabilities.

**Methods**. SD mouse eyes were investigated by light, fluorescence and electron microscopy and were compared to the respective albino and pigmented wild type mice and to a human donor SD eye. To confirm the role of RPE melanin in scavenging oxidative stress, melanin from *S. officinalis* as a standard and porcine RPE were tested for their capability to quench superoxide anions.

**Results**. Histological alterations indicative of oxidative stress and/or lysosomal dysfunction were present in albino *Abca4*$^{-/-}$ and *Abca4*$^{-/-}$.*Rdh8*$^{-/-}$ mice. Retinal damage, such as inner segment rupture and pyknotic or free photoreceptor nuclei in the subretinal space and RPE vacuolization were exclusively found in albino *Abca4*$^{-/-}$ mice. Shortened and disorganized photoreceptor outer segments and dead RPE cells were found in albino *Abca4*$^{-/-}$ and *Abca4*$^{-/-}$.*Rdh8*$^{-/-}$ mice, with earlier onset in albino *Abca4*$^{-/-}$ mice. Undegraded phagosomes and lipofuscin accumulation were present in the RPE of all three SD strains, but numbers were highest in *Abca4*$^{-/-}$.*Rdh8*$^{-/-}$ mice. Lipofuscin morphology differed between SD strains: (melano-)lipofuscin granules in pigmented *Abca4*$^{-/-}$ mice had a homogenous electron density and sharp demarcations, while lipofuscin in albino *Abca4*$^{-/-}$ mice had a flocculent electron density and often lacked a surrounding membrane, indicating loss of lysosomal integrity. Young *Abca4*$^{-/-}$.*Rdh8*$^{-/-}$ mice showed (melano-)lipofuscin granules with homogenous electron density, while in aged animals granules with flocculent electron density predominated. Both strains of pigmented SD mice had melanolipofuscin clusters as found in the human SD eye. Like melanin from *S. officinalis*, porcine RPE melanin can also quench superoxide anions.

Corresponding author
Tatjana Taubitz, tatjana.taubitz@med.uni-tuebingen.de

**Discussion**. The presented pathologies in albino *Abca4*$^{-/-}$ and *Abca4*$^{-/-}$.*Rdh8*$^{-/-}$ mice suggest oxidative stress and/or lysosomal dysfunction within the RPE. Since albino *Abca4*$^{-/-}$ mice have the earliest onset and severest damage and as absence of melanin and also melanin turnover with age are known to diminish RPEs anti-oxidative properties, we assume that RPE melanin plays a role in SD related damages. A lack of pathology in pigmented *Abca4*$^{-/-}$ mice due to lower stress levels as compared to the *Abca4*$^{-/-}$.*Rdh8*$^{-/-}$ mice underlines this hypothesis. It is also supported by the finding that RPE melanin can quench superoxide anions. We therefore suppose that RPE melanin is important in retinal health and we discuss its role as an oxidative stress scavenger.

## INTRODUCTION

Stargardt disease (STGD1, OMIM #248200) is one of the most frequent inherited macular dystrophies in humans and affects 1 in 8,000–10,000 individuals (*Blacharski, 1988*). It is caused by mutations in the ATP-binding cassette A4 (*ABCA4*) gene, which encodes for a transmembrane protein located in the rim of photoreceptor disks (*Allikmets et al., 1997*; *Molday, Zhong & Quazi, 2009*). The ABCA4 protein is involved in all-trans retinal transport through the photoreceptor disk membrane and is thus part of the visual cycle. Disrupted ABCA4 function leads to accumulation of all-trans retinal within the disk and facilitates formation of N-retinylidene-N-retinylethanolamine (A2E) and other bisretinoids (*Sparrow et al., 2012*). These substances are major components of lipofuscin, a cytotoxic age pigment that accumulates in the retinal pigment epithelium (RPE) and leads to RPE dysfunction and subsequent RPE and photoreceptor degeneration (*Sparrow et al., 2012*). Owing to the burden of cytotoxic lipofuscin accumulation early in life, Stargardt patients suffer from early onset macular degeneration, resulting in progressive bilateral vision loss.

The first animal model generated for Stargardt disease was the pigmented *Abca4*$^{-/-}$ mouse (*Weng et al., 1999*). This model shows several typical pathological changes related to Stargardt disease, including accumulation of lipofuscin granules in the RPE, elevated levels of A2E and other bisretinoid fluorophores and delayed dark adaption (*Charbel Issa et al., 2013*). Since this mouse strain lacks retinal degeneration (*Charbel Issa et al., 2013*; *Weng et al., 1999*), it is considered as a model for the early stage of Stargardt disease.

A second *Abca4*$^{-/-}$ mouse strain, the albino *Abca4*$^{-/-}$ mouse, was generated by crossbreeding pigmented *Abca4*$^{-/-}$ mice with Balb/c mice (*Radu et al., 2004*). This albino *Abca4*$^{-/-}$ mouse model differs from the pigmented *Abca4*$^{-/-}$ mouse model in several aspects. Albino *Abca4*$^{-/-}$ mice have lower levels of the lipofuscin component A2E, but higher levels of A2E oxidation products, so called A2E oxiranes than pigmented *Abca4*$^{-/-}$ mice (*Radu et al., 2004*). It was previously shown that lipofuscin can generate reactive oxygen species, including singlet oxygen, after exposure to blue light (*Rozanowska et al., 1995*) and that A2E is oxidized by singlet oxygen to A2E oxiranes *in vitro*

(*Ben-Shabat et al., 2002*). As a consequence, A2E can auto-oxidize to A2E oxiranes in the presence of light. Since A2E oxiranes are highly reactive and can cause DNA fragmentation (*Sparrow et al., 2003a*; *Sparrow, Zhou & Cai, 2003b*), this might be an important mechanism of A2E cytotoxicity. In contrast to pigmented *Abca4*$^{-/-}$ mice, albino *Abca4*$^{-/-}$ mice show a mild retinal degeneration, starting at an age of 7 months (*Radu et al., 2011*) and leading to a loss of approximately 30 to 40% of photoreceptor nuclei at the age of 11 months compared to WT (*Radu et al., 2008*; *Wu, Nagasaki & Sparrow, 2010*). Furthermore, albino *Abca4*$^{-/-}$ mice show a range of signs of chronic inflammation in the RPE including upregulated expression of oxidative stress genes, elevated levels of the oxidative stress markers malondialdehyde and 4-hydroxynonenal, activation of complement, downregulation of complement regulatory proteins and monocyte chemoattractant protein-1, increased C-reactive protein immunoreactivity and thickening of Bruch's Membrane (*Radu et al., 2011*).

A third Stargardt disease mouse model is the pigmented *Abca4*$^{-/-}$.*Rdh8*$^{-/-}$ mouse strain, generated by crossbreeding *Abca4*$^{-/-}$ mice with *Rdh8*$^{-/-}$ mice (*Maeda et al., 2008*). *Abca4*$^{-/-}$.*Rdh8*$^{-/-}$ mice are described to have reduced all-trans retinal clearance and higher amounts of A2E, A2E-precursors and retinal dimer conjugates than WT and pigmented *Abca4*$^{-/-}$ mice (*Maeda et al., 2008*). Regional retinal degeneration and rosette formation is observed by as early as 6 weeks of age and leads to advanced retinal degeneration by the age of 3 months (*Maeda et al., 2008*). Furthermore, drusen formation and thickening of Bruch's membrane in 3-month-old animals and choroidal neovascularization in 10-month-old animals are described (*Maeda et al., 2008*).

So far, ultrastructural changes were not investigated in detail in all three mentioned Stargardt disease mouse models, although they are widely used in the search of therapy options for SD. In this study, we conducted light and fluorescence microscopic as well as ultrastructural analysis of age-related retinal and RPE changes in these mouse strains. For comparison, we investigated tissue of the respective WT mouse strains and a single eye of a human donor with Stargardt disease. So far, electron microscopic examinations of human Stargardt eyes are scarce (*Birnbach et al., 1994*; *Bonilha et al., 2016*; *Eagle Jr et al., 1980*).

We found early onset of histologic alterations typical for oxidative stress and lysosomal impairment in albino *Abca4*$^{-/-}$ mice. Similar alterations were found with later onset and less severity in pigmented *Abca4*$^{-/-}$.*Rdh8*$^{-/-}$ mice and were lacking in pigmented *Abca4*$^{-/-}$ mice. Since both the absence of melanin and melanin ageing are known to diminish the antioxidative properties of the RPE, these finding underline the importance of RPE melanin for retinal health and disease.

# MATERIALS AND METHODS

## Animals

Pigmented *Abca4*$^{-/-}$ mice (129S4/SvJae-Abca4$^{tm1Ght}$) were kindly donated by P. Charbel Issa (University of Oxford, Oxford, England). Albino *Abca4*$^{-/-}$ mice (BALB/c-Abca4 $^{tm1Ght}$) were kindly donated by G. Travis and R. Radu (University of California, Los Angeles, CA). Double knock-out *Abca4*$^{-/-}$.*Rdh8*$^{-/-}$ (129S4/SvJae- Abca4$^{tm1Ght}$*C57BL/6J-Rdh8$^{tm1Kpal}$) mice were kindly donated by K. Palczewski (Case Western Reserve University,

Cleveland, Ohio). Control albino WT mice (Balb/c) were purchased from Charles River (Sulzfeld, Germany) and control pigmented WT mice (129S2) were purchased from Harlan Laboratories (Hillcrest, UK). The knock-out mouse strains were bred in our in-house facility. Light cycling was 12 h light (approximately 50 lux in cages)/12 h dark, food and water were available *ad libitum*. All procedures involving animals were in accordance with the German laws governing the use of experimental animals and were previously approved by the local agency for animal welfare (Einrichtung für Tierschutz, Tierärztlichen Dienst und Labortierkunde der Eberhard Karls Universität Tübingen, Tuebingen, Germany) and the local authorities (Regierungspräsidium Tübingen, Tuebingen, Germany).

## Sample preparation and light, fluorescence and electron microscopy

Animals were sacrificed by carbon dioxide inhalation and subsequent cervical dislocation and the eyes were immediately enucleated.

For light and electron microscopy, eyes were fixed overnight at 4 °C in 5% glutaraldehyde in 0.1 M cacodylate buffer (pH 7.4). After washing in cacodylate buffer, the cornea and lens were removed and eye cups were hemisected. Halves were post-fixed in 1% osmium tetroxide in 0.1 M cacodylate buffer and bloc-stained with uranyl acetate. Samples were dehydrated in a graded series of ethanol and propylene oxide and embedded in Epon. Reagents were purchased from AppliChem (Darmstadt, Germany), Merck (Darmstadt, Germany) and Serva (Heidelberg, Germany). Light microscopy on toluidine blue stained semi-thin sections (500 nm) was performed with a Zeiss Axioskop (Zeiss, Jena, Germany). Ultrathin sections (70 nm) were mounted on copper slot grids (Plano, Wetzlar, Germany) and stained with lead citrate and examined with a Zeiss 900 electron microscope (Zeiss, Jena, Germany).

For fluorescence microscopy and immunohistochemistry, eyes were fixed in 4.5% formaldehyde (Carl Roth, Karlsruhe, Germany) and embedded in paraffin wax, cut into 5 μm thick sections for fluorescence microscopy and 4 μm thick sections for immunohistochemistry. Samples for fluorescence microscopy were deparaffinized according to standard procedures, cover-slipped with FluorSave (Calbiochem, La Jolla, CA, USA) and were investigated with a Zeiss Axioplan2 imaging microscope (Zeiss, Jena, Germany). Lipofuscin autofluorescence was visualized with a custom lipofuscin filter set (excitation 360 nm, emission 540 nm).

## Immunohistochemistry

Sections were deparaffinized, rehydrated and subjected to heat induced antigen retrieval in a pressure cooker for 2 min in either citrate buffer at pH 6 (anti-HNE, anti-MDA) or Tris buffer at pH 9 (anti-NITT). Sections were incubated overnight at 4 °C with either rabbit anti-HNE (4-hydroxy-2-noneal, HNE11-S, alpha diagnostic international; major product of endogenous lipid peroxidation, 1:3,500), rabbit anti-MDA (malondialdehyde, MDA11-S, alpha diagnostic international; byproduct of endogenous lipid peroxidation, 1:2,000) or rabbit anti-NITT (nitrotyrosine, NITT12-A, alpha diagnostic international; peroxynitrite-induced nitration of tyrosine residues in proteins, 1:1,000). Dilution of primary antibodies was done with antibody diluent with background-reducing components

(Dako S3022; Agilent, Santa Clara, CA, USA). The primary antibodies were detected with the Dako REAL$^{TM}$ Detection System, Alkaline Phosphatase/RED, Rabbit/Mouse according to the manufacturer's instructions. In brief, sections were incubated with biotinylated goat anti-mouse and anti-rabbit immunoglobulins for 15 min at room temperature with subsequent incubation with streptavidin conjugated to alkaline phosphatase for 15 min at room temperature. Reaction products were visualized with freshly prepared substrate working solution supplemented with levamisole to block endogenous peroxidases. Sections were counterstained with hematoxylin solution modified according to Gill III (Merck, Darmstadt, Germany). To control for non-specific binding, control sections were prepared without incubation with primary antibodies. Sections were investigated using a Zeiss Axioskop (Zeiss, Jena, Germany).

## Quantification of photoreceptor nuclei and outer segment length by light microscopy

Semi-thin sections of whole eye cups (up to 400 µm from the optic nerve head) were photographed using a 63× oil objective. Areas adjacent to the optic nerve and to the ora serrata were excluded from analysis due to the physiological thinning of the retina.

Photoreceptor nuclei were counted semi-automatically in an average of 11 digital images per section in on average three eyes per age group using Fiji software (*Schindelin et al., 2012*). In each image, a region of interest with a width of 100 µm along the outer nuclear layer was defined and total photoreceptor nuclei within the region of interest were counted as number of photoreceptor nuclei per 100 µm width of retina. The total numbers of eyes investigated by light microscopy and used for photoreceptor nuclei quantification per strain were 34 eyes (albino *Abca4*$^{-/-}$ mice), 30 eyes (pigmented *Abca4*$^{-/-}$ mice) and 47 eyes (*Abca4*$^{-/-}$.*Rdh8*$^{-/-}$ mice).

Outer segment layer length was measured manually in 11–16 positions per section in an average of two eyes per age group using Fiji software.

## Electron microscopic investigation

Whole sections of mouse eyes were thoroughly investigated by electron microscopy. Total numbers of eyes investigated by electron microscopy for the different strains were 12 eyes (pigmented *Abca4*$^{-/-}$), 16 eyes (albino *Abca4*$^{-/-}$), 16 eyes (*Abca4*$^{-/-}$.*Rdh8*$^{-/-}$), six eyes (albino WT) and three eyes (pigmented WT). Sections were examined for changes in photoreceptor, RPE and Bruch's membrane structure. Lipid droplets in RPE cells were counted and their diameter measured in whole sections. Thickness of Bruch's membrane was measured in 15 consecutive images. All measurements were performed with iTEM 5.0 Software (Olympus Soft Imaging Solutions, Muenster; Germany).

## Quantification of the area in RPE cytoplasm occupied by lipofuscin-like material

Lipofuscin-like material (derived from both lipofuscin and melanolipofuscin granules, hereafter referred to as lipofuscin) was quantified in young (aged 4 to 6 months) and old (aged 12 months) Stargardt mice in an average of 5 eyes per group. Thirty electron micrographs of the RPE (magnification× 20.000) per eye were taken in a way so only

cytoplasm and no nuclei, microvilli and basal labyrinth were visible. To compensate for vignetting, only an oval region of interest in the image center was analyzed. The Trainable Weka Segmentation (*Arganda-Carreras et al., 2017*) plugin for Fiji software, a machine learning tool, was used to determine the area of lipofuscin per image. A representative subset of images was used to train the tool to specifically recognize lipofuscin, so melanin was excluded from the analysis. A combination of the mean, median, maximum, minimum, variance and the anisotropic diffusion-, entropy- and neighbor algorithms was used. Segmentation calculations were performed on a 16 virtual central processing units system with 64 GB working memory (de.NBI Cloud Tübingen, https://denbi.uni-tuebingen.de).

## Human tissue

Glutaraldehyde-fixed perimacular tissue of a 72-year-old donor with clinically diagnosed Stargardt disease was obtained through Foundation Fighting Blindness (Columbia, MD, USA). Post mortem time until fixation was 48 h. Written informed consent of the donor for use in medical research and approval of the Institutional Review Board of the University of Tuebingen (Ethik-Kommission an der Medizinischen Fakultät der Eberhard-Karls-Universität und am Universitätsklinikum Tübingen, Tuebingen, Germany, approval number 462/2009BO2) were obtained. The experiments were performed in adherence to the tenets of the Declaration of Helsinki. Human tissue was embedded for standard electron microscopy as described above. For investigation of lipofuscin autofluorescence in semi-thin sections, heavy-metal treatment during embedding (osmium tetroxide, uranyl acetate) was omitted.

## Isolation of porcine RPE melanin

RPE melanin from pig eyes was isolated as described by (*Boulton & Marshall, 1985*) with modifications (*Zareba et al., 2006*). In brief, pig eyes were opened close to the ora serrata and the anterior segment and vitreous were discarded. The retina was removed with forceps and the eye cup washed with PBS (Gibco, Carlsbad, CA, USA). Eye cups were filled with trypsin/EDTA (Gibco, Carlsbad, CA, USA) and incubated for 10 min at 37 °C. RPE cells were isolated by pipetting jet streams onto the cell layer and collected in DMEM supplemented with 10% fetal bovine serum (Gibco, Carlsbad, CA, USA) to stop the enzymatic reaction. RPE cell pellets were homogenized using a tissue grinder. Cell debris were removed by centrifugation for 7 min at 60 g and the obtained supernatant was centrifuged for 10 min at 6,000 g to pellet all pigment granules. The pellet was collected in 0.3 M sucrose, loaded on a two-step sucrose gradient (1 M and 2 M) and centrifuged for 1 h at 103.000 g. Isolated RPE melanin granules were washed in PBS and the final granule concentration was determined using a hemocytometer.

## Quenching of superoxide in an NBT assay

Quenching capability of melanin was investigated with the photochemical nitroblue tetrazolium (NBT) assay (*Cheng et al., 2015*). In this assay, superoxide generated by illumination of riboflavin reduces pale-yellow NBT to blue-violet formazan. The reaction mixture had a final concentration of 0.1 mM EDTA, 13 μM methionine, 75 μM NBT and 4 mM riboflavin in PBS pH 7.4. EDTA, methionine, NBT and riboflavin were purchased

from Sigma-Aldrich (St. Louis, MO, USA). The reaction mixture was illuminated for 15 min with blue light from an LED lamp at 2.5 mW/cm$^2$ (450 nm, SunaEco 1500 Ocean Blue XP, Tropic Marin, Wartenberg, Germany). Uniform illumination of the samples was ensured by placing the samples on a horizontally aligned rotator. The formazan was quantified at 560 nm (Synergy HT, BioTek, Winooski, VT, USA).

Controls included positive controls (all reagents, no melanin, illuminated), negative controls (all reagents, no melanin, kept in the dark) and blanks (no riboflavin, melanin, illuminated) to compensate for melanin absorption at readout.

### Statistical analysis

Statistical analysis was performed with JMP 13 (SAS, Cary, NC, USA). All data sets were tested for normal distribution to decide on using parametric or non-parametric testing. Since all data sets were not normally distributed and contained several groups, we used the Steel-Dwass All Pairs test, a multiple non-parametrical test with alpha correction. The null hypothesis was that the groups were not significantly different. Values are given as mean ±standard deviation, $p < 0.05$ was considered statistically significant.

## RESULTS

### Photoreceptors

Light microscopic investigation of tissue of 12-month-old animals (Fig. 1A) showed overall typical histology in pigmented *Abca4*$^{-/-}$ mice compared to pigmented WT as already published in (*Charbel Issa et al., 2013*) (not shown). Albino *Abca4*$^{-/-}$ mice had shortened outer segments whereas *Abca4*$^{-/-}$.*Rdh8*$^{-/-}$ showed a hypertrophy of the RPE (Fig. 1A, Fig. S1).

Outer segment length was measured in light microscopy. Compared to age-matched albino WT animals, albino *Abca4*$^{-/-}$ mice aged 4 months showed a significant reduction in outer segment length (albino *Abca4*$^{-/-}$ 4 months 22.3 ± 3.2 μm, albino WT 4 months 26.2 ± 2.5 μm, $p < 0.001$, Fig. 1B). In *Abca4*$^{-/-}$.*Rdh8*$^{-/-}$ mice, outer segment shortening had later onset: no difference was found for 3-month-old animals, whereas 6-month-old animals had shortened outer segments (*Abca4*$^{-/-}$.*Rdh8*$^{-/-}$ 3-month-old 32.1 ± 2.7 μm, pigmented WT 3-month-old 32.6 ± 1.6 μm, not significant; *Abca4*$^{-/-}$.*Rdh8*$^{-/-}$ 6-month-old 27.7 ± 2.0 μm, pigmented WT 12-month-old 33.9 ± 2.0 μm, $p < 0.0001$, Fig. 1B). Pigmented *Abca4*$^{-/-}$ had no outer segment shortening compared to pigmented WT.

A detailed analysis of the age-dependent reduction of photoreceptor nuclei numbers of the three different Stargardt mouse strains revealed similar retinal degeneration courses for both albino *Abca4*$^{-/-}$ and *Abca4*$^{-/-}$.*Rdh8*$^{-/-}$ mice (Fig. 1C) as compared to the pigmented *Abca4*$^{-/-}$ mice for which it is known that they do not present signs of retinal degeneration (*Charbel Issa et al., 2013*; *Weng et al., 1999*). We quantified total photoreceptor nuclei numbers, as opposed to the more commonly used outer nuclear layer thickness, to account for nuclei disorganization and gaps between nuclei that occurred in albino *Abca4*$^{-/-}$ and, to a lesser extent, in *Abca4*$^{-/-}$.*Rdh8*$^{-/-}$ mice.

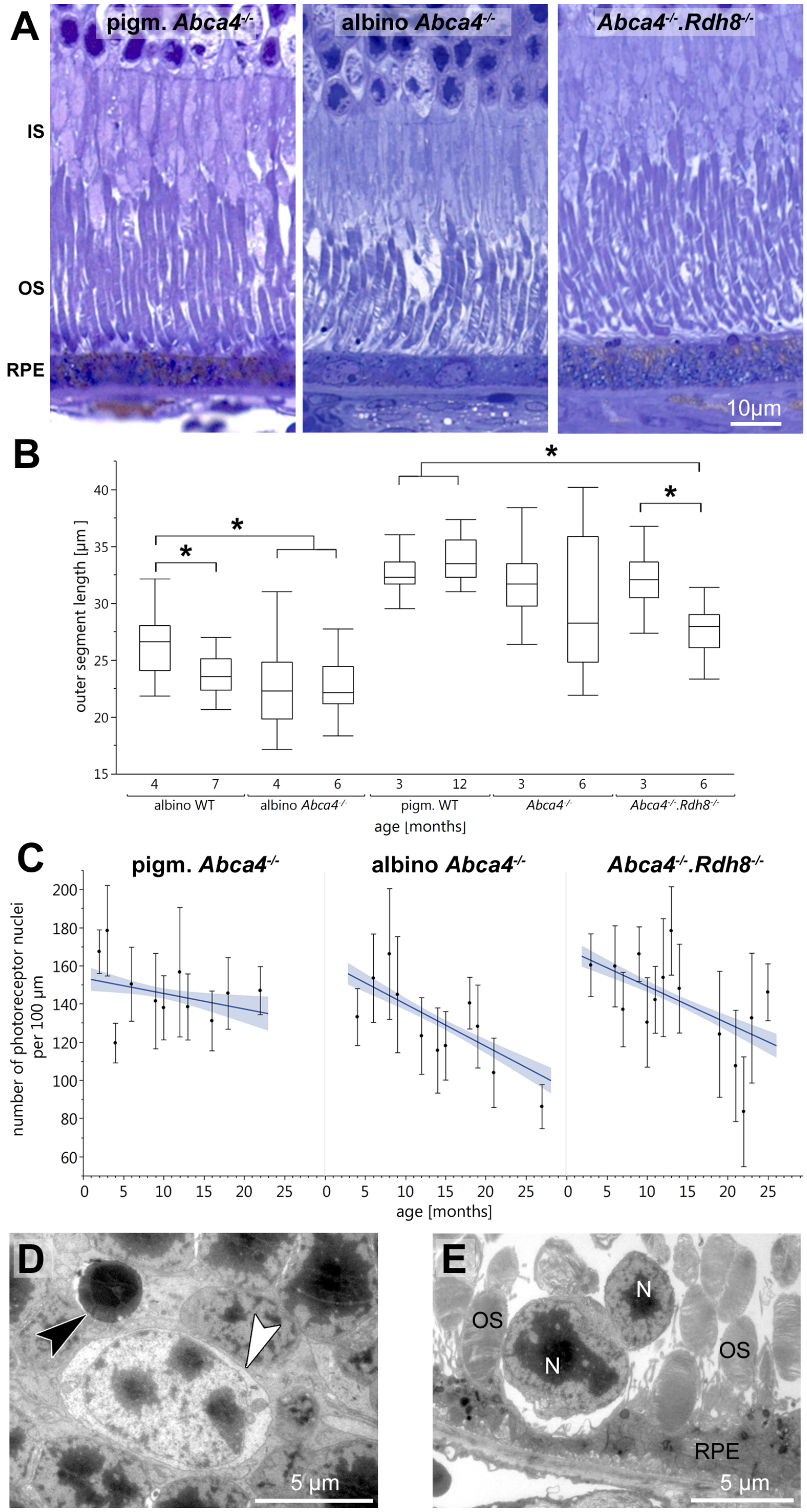

**Figure 1** **Retinal degeneration in Stargardt mice.** (A) Representative light micrographs of 12-month-old pigmented and albino *Abca4*$^{-/-}$ and *Abca4*$^{-/-}$.*Rdh8*$^{-/-}$ mice. Albino *Abca4*$^{-/-}$ mice have shorter outer segments whereas *Abca4*$^{-/-}$.*Rdh8*$^{-/-}$ mice have hypertrophic RPE. (B) Quantification of outer segment length in Stargardt mice compared to WT animals. In albino *Abca4*$^{-/-}$ mice, outer segment length is already reduced in 4-month-old animals, compared to albino WT. 

**Figure 1 (…continued)**
*Abca4*$^{-/-}$.*Rdh8*$^{-/-}$ mice show a significant reduction of outer segment length between 3 and 6 months ($n =$ on average 2 eyes/age group, $*p \leq 0.001$, Steel-Dwass All Pairs test). (C) Quantification of numbers of photoreceptor nuclei per 100 µm width of retina. Linear regression and 95% confidence interval are shown ($n =$ on average 3 eyes/age group). (D) A pyknotic nucleus is present (black arrowhead) next to a cone nucleus (white arrowhead) (albino *Abca4*$^{-/-}$, 4 months). (E) Photoreceptor nuclei are located next to unusually thinned RPE cells (albino *Abca4*$^{-/-}$, 4 months). IS, inner segments; OS, outer segments; RPE, retinal pigment epithelium; N, nucleus.

On the ultrastructural level, shrunken, electron dense nuclei, resembling pyknosis, were only found in albino *Abca4*$^{-/-}$ animals (Fig. 1D). Furthermore, isolated photoreceptor nuclei were present in the subretinal space next to unusually thinned RPE cells, even in young albino *Abca4*$^{-/-}$ animals (Fig. 1E). In our colony of *Abca4*$^{-/-}$.*Rdh8*$^{-/-}$mice, we could not find the previously described rosette formation at 3 weeks and the rather prominent retinal degeneration beginning at 3 months (*Maeda et al., 2008*). Rosette formation was only seen in a single 12-month-old animal. Instead, RPE hypertrophy was regularly seen in *Abca4*$^{-/-}$.*Rdh8*$^{-/-}$animals (Fig. 1A, Fig. S1).

Genotype and age-dependent alterations of the disc membrane stacking were apparent on the ultrastructural level (Fig. 2A, Table 1). Those alterations seemed to start at the outer segment tips (next to the RPE) and spread towards the base of the outer segment. Alterations in outer segment tip organization were never seen in pigmented WT and pigmented *Abca4*$^{-/-}$ mice. In albino WT and *Abca4*$^{-/-}$.*Rdh8*$^{-/-}$ animals, it was not seen in the youngest animals investigated (3–4 months), but in animals aged 6–7 months and older. In albino *Abca4*$^{-/-}$ mice, outer segment tip disorganization was already present in 4-month-old animals, the youngest group investigated in this strain.

In albino *Abca4*$^{-/-}$ and *Abca4*$^{-/-}$.*Rdh8*$^{-/-}$ animals, alterations in overall disc membrane stacking were observed starting with 6 months but were lacking in pigmented and albino WT and pigmented *Abca4*$^{-/-}$ animals.

Inner segments of photoreceptors appeared swollen and/or ruptured in albino *Abca4*$^{-/-}$ mice (Fig. 2B). In pigmented *Abca4*$^{-/-}$ mice and *Abca4*$^{-/-}$.*Rdh8*$^{-/-}$ mice, inner segment damages were only occasionally observed (Table 1).

In albino *Abca4*$^{-/-}$ mice, agglomerated electron dense material was occasionally present between inner segment and outer segment (Fig. 2C) and disk membrane renewal seemed to be somewhat disturbed as disk membrane swirls were sporadically present at the nascent part of the outer segment (Fig. 2D). These changes were not present in albino WT and pigmented Stargardt mice. Outer segment tips were in contact to RPE microvilli in all mouse strains and age groups.

## Vacuole-like structures within the RPE

In semi-thin sections of albino *Abca4*$^{-/-}$ mice, vacuole-like structures within the RPE monolayer were apparent in animals as young as 6 months (Fig. 3A). Electron microscopic investigation revealed that the vacuole-like structures were an enlargement of the intercellular space between RPE cells and were apically limited by the junctional complex (Fig. 3B). Sometimes vacuole-like structures were also seen within the cytoplasm, usually in proximity to the lateral plasma membrane and probably also of intercellular

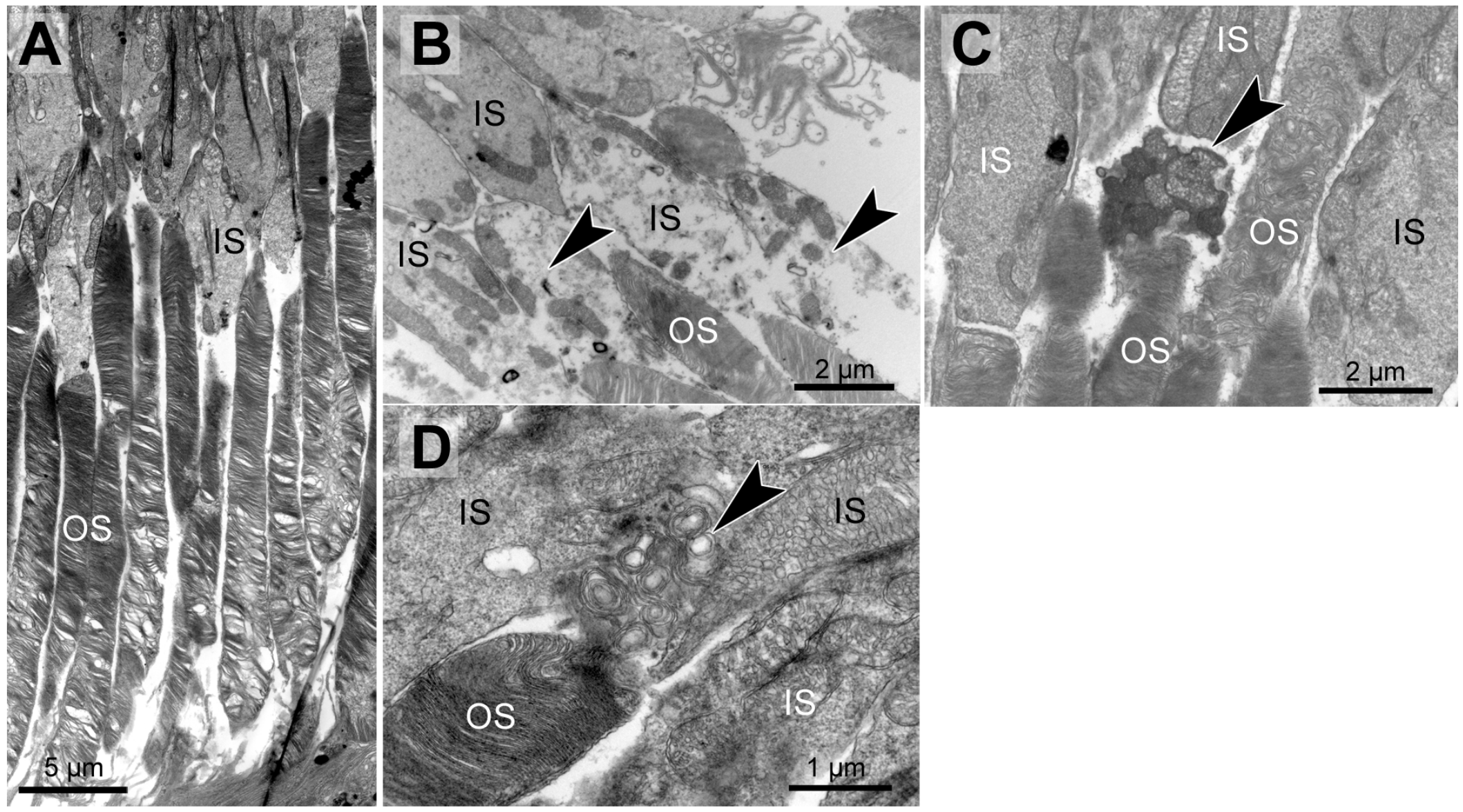

**Figure 2 Morphological changes in inner and outer segments of albino *Abca4*$^{-/-}$ mice.** (A) Disc membrane stacking in outer segments gets more disorganized towards the RPE (albino *Abca4*$^{-/-}$, 6 months). (B) Inner segments are ruptured (arrowheads) (albino *Abca4*$^{-/-}$, 4 months). (C) Electron dense agglomerations (arrowhead) are seen between inner and outer segment (albino *Abca4*$^{-/-}$, 6 months). (D) Newly synthesized disk membranes are disorganized and accumulate outside of the outer segment (arrowhead) (albino *Abca4*$^{-/-}$, 9 months). IS, inner segments; OS, outer segments.

space origin. The contents of the vacuole-like structures varied between empty (not shown), filled with membranous material (Fig. 3B) and filled with a variety of fibrillous, granular, lamellar and amorphous materials (Fig. 3C). Different types of contents were often seen within the same eye. Ultrastructurally, vacuole-like structures were apparent in albino *Abca4*$^{-/-}$ mice of all age groups, but were barely present in pigmented *Abca4*$^{-/-}$ and *Abca4*$^{-/-}$.*Rdh8*$^{-/-}$ mice (Fig. 3D). The number of vacuole-like structures increased with age, though there was a considerable variance between eyes of the same age. Vacuole-like structures were considerably fewer in albino WT mice.

## Lipofuscin

To compare the lipofuscin autofluorescence between the different Stargardt mouse strains, paraffin sections of 12-month-old animals were investigated by fluorescence microscopy (Fig. 4A). In pigmented *Abca4*$^{-/-}$ sections, the lipofuscin autofluorescence was clearly defined, indicating the presence of granules with well-defined borders. In albino *Abca4*$^{-/-}$ sections, the lipofuscin autofluorescence was predominantly blurry. In *Abca4*$^{-/-}$.*Rdh8*$^{-/-}$ sections, both clearly defined and blurry lipofuscin autofluorescence were present. Autofluorescence in albino *Abca4*$^{-/-}$ and *Abca4*$^{-/-}$.*Rdh8*$^{-/-}$ tissue was elevated compared to pigmented *Abca4*$^{-/-}$ tissue.

The morphological appearance of lipofuscin granules in the different Stargardt mouse strains was investigated by electron microscopy. Typical morphological appearances are

**Table 1** **Ultrastructural observations in WT and Stargardt mice.**

| | albino WT | | albino *Abca4*$^{-/-}$ | | | | | | | pigm. WT | | pigm. *Abca4*$^{-/-}$ | | | | | | | | *Abca4*$^{-/-}$.*Rdh8*$^{-/-}$ | | | | | | | |
|---|---|---|---|---|---|---|---|---|---|---|---|---|---|---|---|---|---|---|---|---|---|---|---|---|---|---|---|
| **Age [months]** | **4** | **7** | **4** | **6** | **9** | **12** | **15** | **18** | **21** | **3** | **12** | **2** | **4** | **6** | **9** | **12** | **16** | **18** | **22** | **3** | **6** | **9** | **12** | **14** | **19** | **23** | **25** |
| OS tips disarranged | – | +/– | +/– | + | + | + | + | + | + | – | – | – | – | – | – | – | – | – | – | – | +/– | +/– | + | +/– | +/– | +/– | +/– |
| OS on full length disarranged | – | – | – | +/– | + | + | + | + | + | – | – | – | – | – | – | – | – | – | – | – | +/– | +/– | + | + | + | + | + |
| Swollen/ ruptured IS | – | – | + | + | + | + | + | + | + | – | – | – | – | – | +/– | – | – | – | – | – | – | – | +/– | – | – | – | – |
| Lipofuscin: | | | | | | | | | | | | | | | | | | | | | | | | | | | |
| – homogenous | +/– | +/– | + | + | +/– | + | + | + | + | +/– | + | + | ++ | ++ | +++ | ++++ | ++++ | +++ | +++ | + | ++ | + | ++++ | +++ | +++ | + | ++ |
| – flocculent | +/– | +/– | + | ++ | +++ | +++ | +++ | +++ | +++ | +/– | +/– | – | – | – | – | – | – | – | – | – | – | – | +/– | +++ | ++ | ++ | ++ |
| Lipid droplets: | | | | | | | | | | | | | | | | | | | | | | | | | | | |
| – apical | ++ | ++ | +/– | – | – | – | +/– | – | – | + | ++ | +/– | – | – | – | – | – | – | – | – | – | – | – | – | – | – | – |
| – basal | – | ++ | + | ++ | ++ | + | + | + | +/– | + | ++ | + | +/– | +/– | + | +/– | – | +/– | – | +/– | + | +/– | – | +/– | +/– | +/– | +/– |
| Detached RPE cells | +/– | + | + | + | +/– | + | + | +/– | +/– | – | – | – | – | – | – | – | - | +/– | – | – | – | – | – | +/– | + | + | + |
| Undegraded phagosomes | – | + | + | +/– | + | + | + | ++ | + | +/– | + | + | + | + | +/– | + | + | + | + | + | ++ | +++ | + | + | + | + | ++ |
| Lateral basal labyrinth expansions | +/– | + | + | + | + | + | + | + | + | – | – | – | – | – | – | – | – | – | – | – | – | – | – | – | – | + | – |

**Notes.**

–, not or rarely observed; +/–, occasionally observed; +, regularly observed; ++, regularly observed in high amounts; +++, regularly observed in extreme amounts.

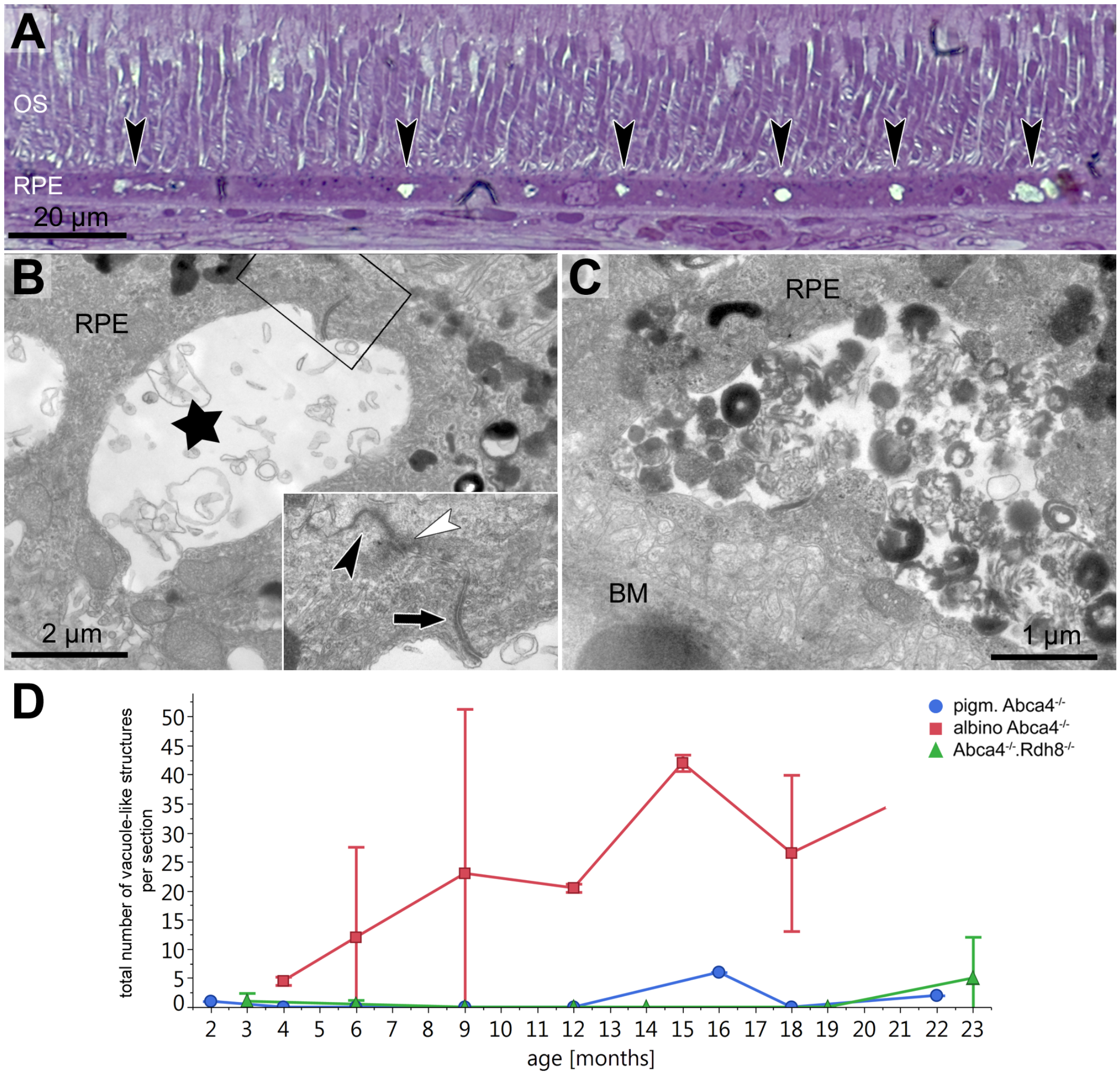

**Figure 3 Vacuole-like structures in RPE cells are most prominent in albino *Abca4*$^{-/-}$ mice.** (A) In light microscopic images, vacuole-like structures (black arrowheads) within the RPE layer are present in albino *Abca4*$^{-/}$ mice (6 months). (B) A vacuole-like structure (asterisk) turns out to be an enlargement of the intercellular space. Inset: apically, the vacuole-like structure is limited by a junctional complex consisting of a desmosome (arrow), zonula adherens (white arrowhead) and tight junction (black arrowhead) (albino *Abca4*$^{-/-}$, 9 months). (C) Vacuole-like structures are regularly filled with very heterogeneous types of material and are surrounded by a membrane (albino *Abca4*$^{-/-}$, 15 months). (D) Quantification in electron microscopy revealed that vacuole-like structures are barely present in pigmented *Abca4*$^{-/-}$ and *Abca4*$^{-/-}$.*Rdh8*$^{-/-}$ mice ($n =$ on average 2 eyes/age group). OS, outer segments; RPE, retinal pigment epithelium; BM, Bruch's membrane.

illustrated in Fig. 4B and age-dependent quantification of the different identified lipofuscin subtypes can be found in Table 1.

In pigmented *Abca4*$^{-/-}$ mice, we found exclusively lipofuscin with homogenous electron density and clearly defined margins indicating an intact limiting membrane (Fig. 4B), as previously described (*Charbel Issa et al., 2013*). Most of the lipofuscin was fused to melanin and formed melanolipofuscin.

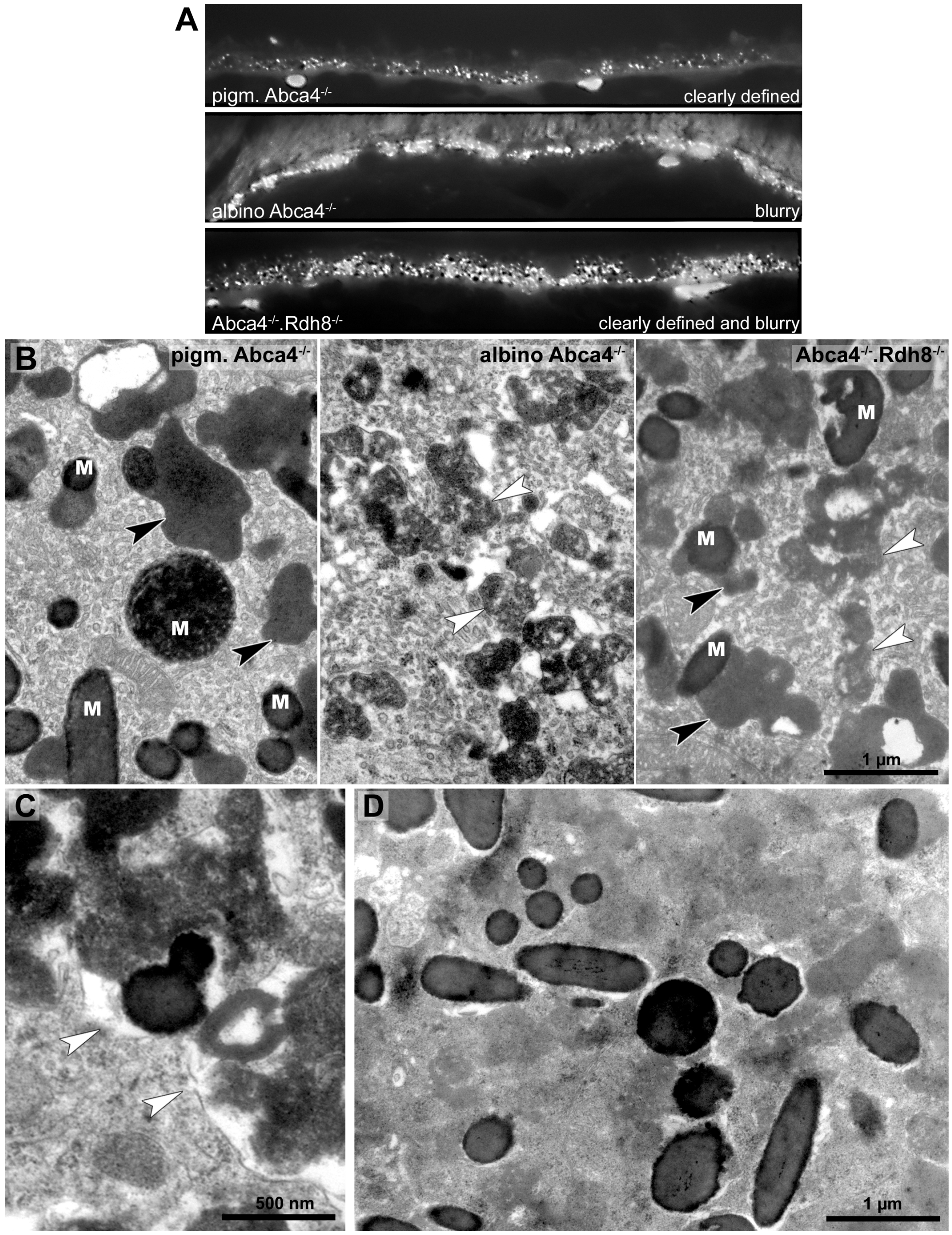

**Figure 4 Lipofuscin in Stargardt mice is highly variable in fluorescence and electron microscopy.** (A) Comparison of lipofuscin autofluorescence in paraffin-embedded sections of 12-month-old Stargardt mice. (B) Lipofuscin and melanolipofuscin have different morphologies in the different Stargardt mouse strains. In pigmented *Abca4*$^{-/-}$ mice, lipofuscin has a homogenous electron-density and sharp demarcations (black arrowheads), while in albino *Abca4*$^{-/-}$ mice, most lipofuscin granules have a flocculent electron-density with unclear demarcations (white arrowheads). While in young *Abca4*$^{-/-}$.*Rdh8*$^{-/-}$ mice, only granules with homogenous electron density can be found, both lipofuscin morphologies can be found in aged animals. M: melanin granules (pigmented and albino *Abca4*$^{-/-}$, 9 months, *Abca4*$^{-/-}$.*Rdh8* $^{-/-}$, 12 months). (C) Membranes surrounding lipofuscin and melanolipofuscin with flocculent electron density are often damaged (white arrowheads) (*Abca4*$^{-/-}$.*Rdh8*$^{-/-}$, 21 months). (D) In very old Stargardt mice, the RPE cytoplasm is abnormally electron dense. Borders of individual lipofuscin granules are barely visible (*Abca4*$^{-/-}$.*Rdh8* $^{-/-}$, 23 months).

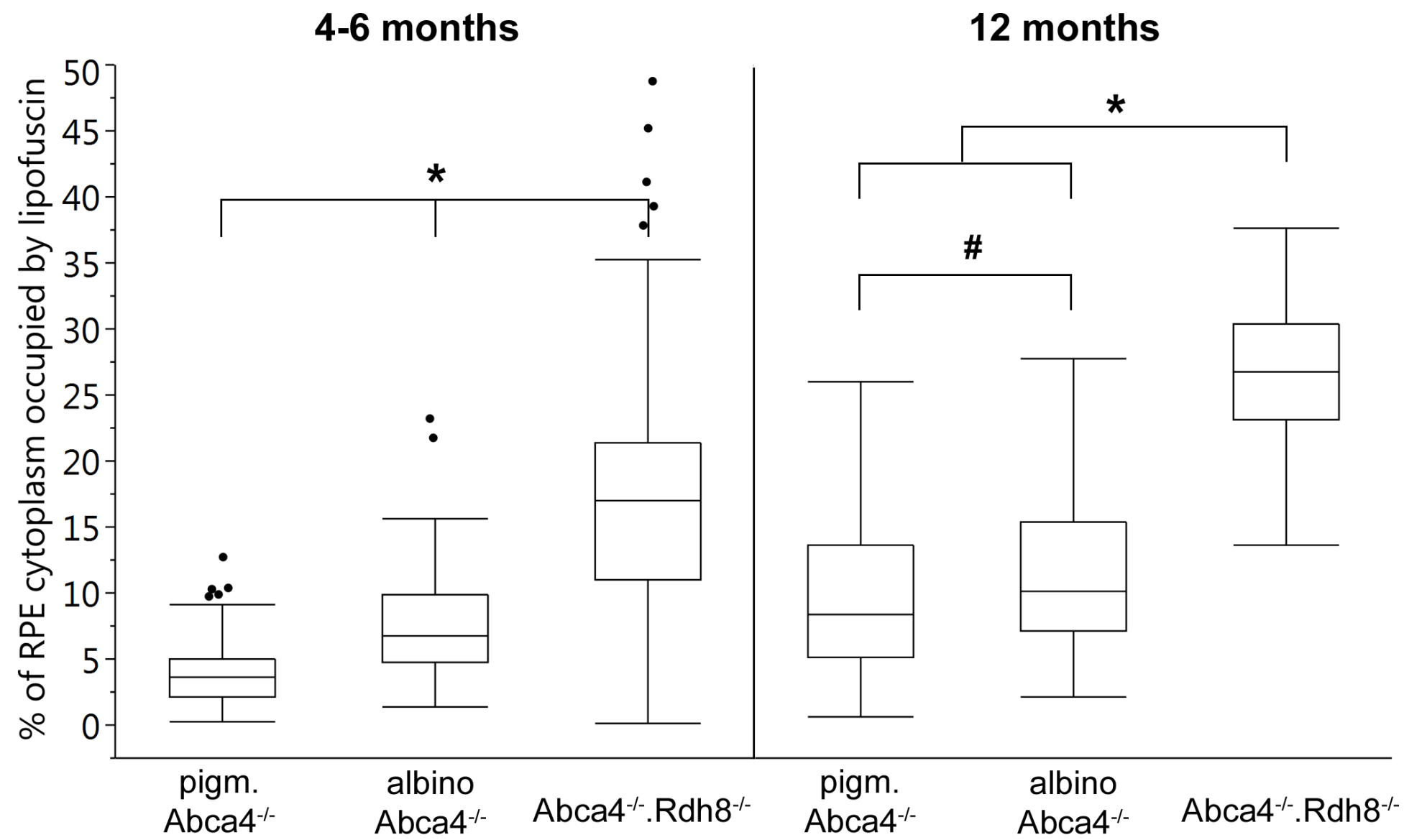

**Figure 5 Quantification of the RPE area occupied by lipofuscin in young and old Stargardt mouse strains.** Lipofuscin granules and lipofuscin moieties of melanolipofuscin granules were taken into account. ($n$ = on average 5 eyes/age group, # $p < 0.05$, * $p < 0.0001$, Steel-Dwass All Pairs test).

In albino *Abca4*$^{-/-}$ mice, lipofuscin with homogenous electron density was also seen. However, the majority of lipofuscin granules belonged to another type that had a flocculent electron density (Fig. 4B, Table 1). The margins of this type of lipofuscin were often not clearly defined, suggesting loss of the lysosomal membrane. As albino *Abca4*$^{-/-}$ mice accumulated lipofuscin with age, the amount of lipofuscin with flocculent electron density increased while the amount of lipofuscin with homogenous electron density seemed to stay constant (Table 1).

In *Abca4*$^{-/-}$.*Rdh8*$^{-/-}$ mice, the first type of lipofuscin with homogenous electron density had a considerable variability between age groups (Table 1). After the age of 12 months, the second type of lipofuscin with flocculent electron density was also present (Table 1). As in pigmented *Abca4*$^{-/-}$ mice, most of the lipofuscin was fused to melanin and formed melanolipofuscin.

Membranes enclosing lipofuscin of the flocculent electron density type were often damaged or absent (Fig. 4C). In very old animals (aged 23 months and older), the RPE cytoplasm was abnormally electron dense (Fig. 4D). Clear demarcations between lipofuscin granules and the cytoplasm were hardly visible.

The area of RPE cytoplasm occupied by lipofuscin (both derived from lipofuscin and melanolipofuscin granules) was quantified in electron micrographs of young (4–6 months) and old (12 months) Stargardt mice (Fig. 5). In both age groups, *Abca4*$^{-/-}$.*Rdh8*$^{-/-}$ mice had the highest levels of lipofuscin (young 16.9% ± 8.3%, old 26.9% ± 5.3%). In young animals, albino *Abca4*$^{-/-}$ mice had almost twice the levels of pigmented *Abca4*$^{-/-}$ mice (albino *Abca4*$^{-/-}$ 7.5% ± 3.9%, pigmented *Abca4*$^{-/-}$ 3.9% ± 2.7%, $p < 0.0001$). In 12-month-old animals, the difference of lipofuscin levels in albino and pigmented

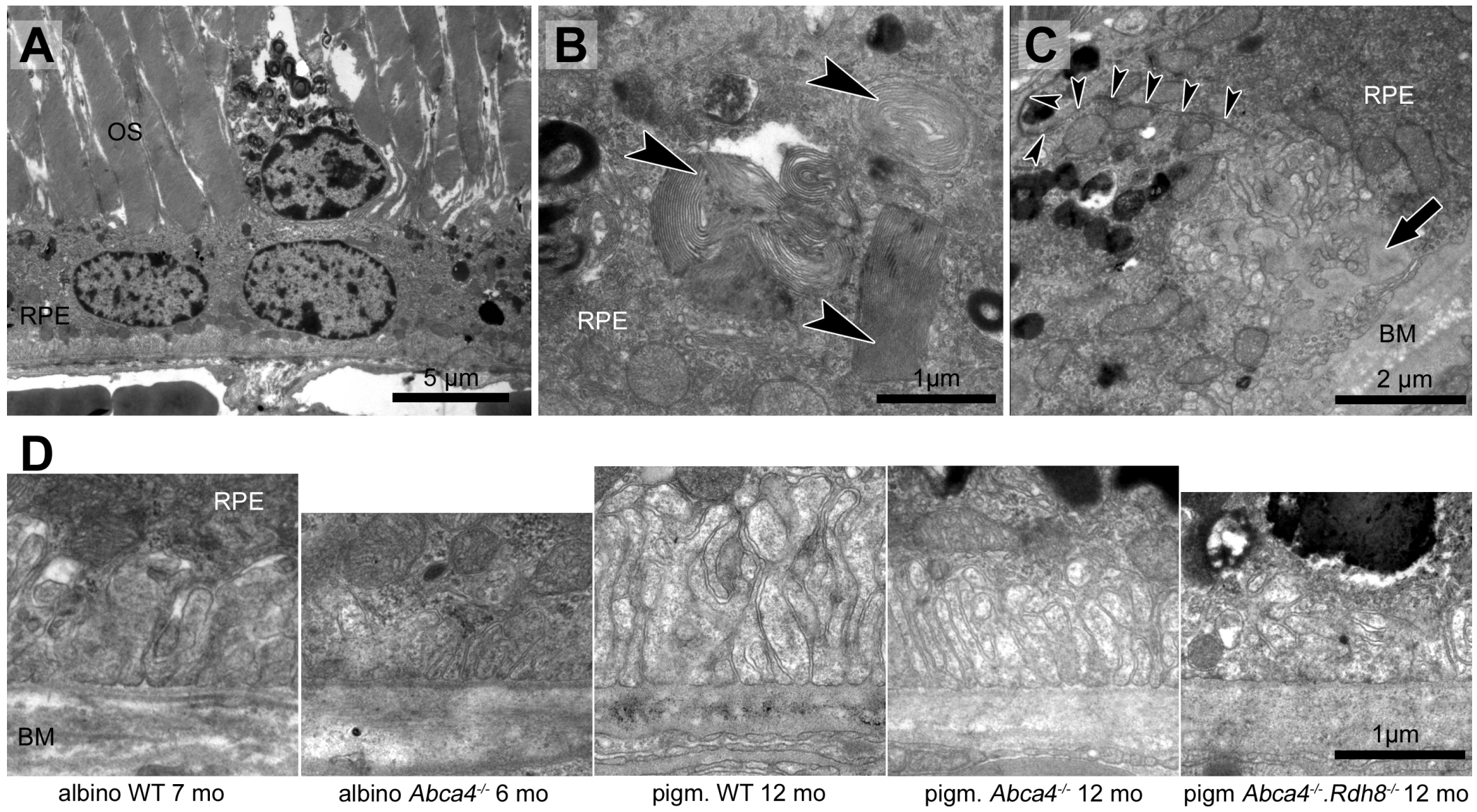

**Figure 6 RPE integrity is compromised in albino *Abca4*$^{-/-}$ mice.** (A) Detached and lysed RPE cell situated in the subretinal space (albino *Abca4*$^{-/-}$, 4 months). (B) RPE cells of all Stargardt mouse strains regularly contain undigested outer segments in the basal part. Few cells in albino *Abca4*$^{-/-}$ mice contain several large undigested outer segments (arrowheads) that claimed the majority of the cytoplasm (albino *Abca4*$^{-/-}$, 4 months). (C) Expansion of the basal labyrinth along the lateral cell borders is regularly seen in albino *Abca4*$^{-/-}$ animals (9 months). Arrowheads indicate the cell border. Basal infoldings were filled with a homogenous material (arrow). (D) Basal labyrinth expansion is reduced in the Stargardt mouse strains compared to the respective wild types. OS, outer segments; RPE, retinal pigment epithelium; BM, Bruch's membrane.

*Abca4*$^{-/-}$ mice was lower but nevertheless statistically significant (albino *Abca4*$^{-/-}$ 11.0% ± 5.5%, pigmented *Abca4*$^{-/-}$ 9.3% ± 5.5%, $p < 0.05$).

## Lipid droplets

Lipid droplets were frequently seen in RPE cells of all investigated mouse strains, especially in albino *Abca4*$^{-/-}$ mice (Table 1). In pigmented and albino WT animals, lipid droplets were small (median diameter: albino WT 235 nm, pigmented WT 353 nm) and the majority of droplets was fused to lipofuscin (albino WT 59% and pigmented WT 82% of total lipid droplets, Fig. S2). They were found in both apical and basal parts of the RPE cells, with a tendency to be more often apically located (Table 1). In Stargardt mouse strains, lipid droplets were bigger (median diameter: pigmented *Abca4*$^{-/-}$ 555 nm, albino *Abca4*$^{-/-}$ 705 nm, *Abca4*$^{-/-}$.*Rdh8*$^{-/-}$ 687 nm) and only sporadically seen in the apical part of the RPE (Table 1, Fig. S2). Fusion with lipofuscin was rarely seen in albino *Abca4*$^{-/-}$ mice (3% of total lipid droplets) and absent in pigmented Stargardt mice.

## RPE integrity and Bruch's membrane

Dead RPE cells were regularly encountered in eyes of albino *Abca4*$^{-/-}$ mice of all ages (Fig. 6A, Table 1). In contrast, dead RPE cells were only found in aged *Abca4*$^{-/-}$.*Rdh8*$^{-/-}$ mice (12 months and older). In pigmented *Abca4*$^{-/-}$ and young *Abca4*$^{-/-}$.*Rdh8*$^{-/-}$ mice,

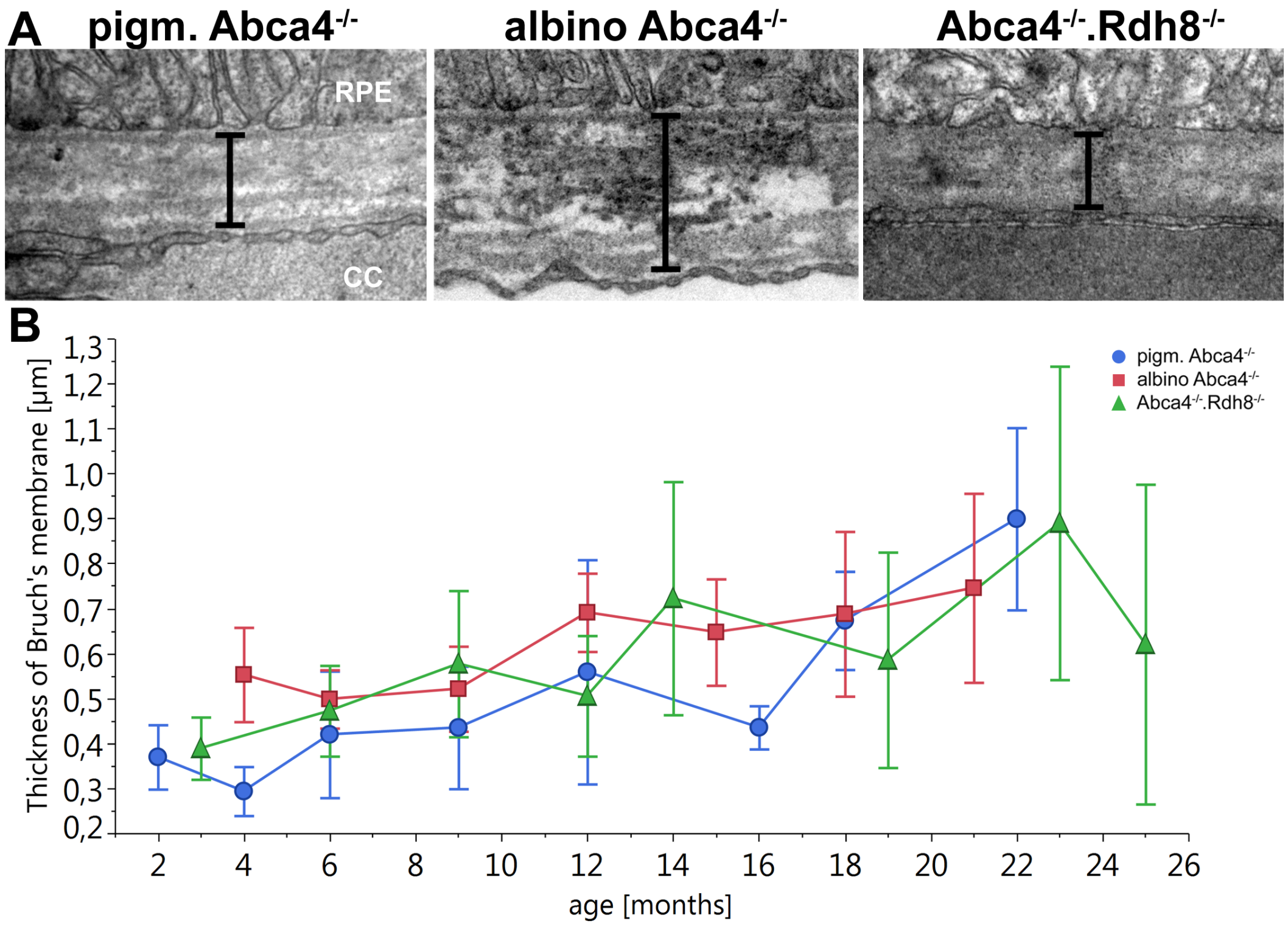

**Figure 7 Thickening of Bruch's Membrane is comparable in all Stargardt mouse strains.** (A) Representative electron microscopic images of Bruch's membrane in 12-month-old Stargardt mice. Thickness of Bruch's membrane was measured from the basement membrane of the RPE to the basement membrane of the endothelial cell of the choriocapillaris (indicated by bars). (B) Thickness of Bruch's membrane increases in all Stargardt mice to the same extent. There is no significant difference between the different Stargardt mouse strains. ($n$ = on average 2 eyes/age group) RPE, retinal pigment epithelium; CC, choriocapillaris.

dead RPE cells were not found. Locally, RPE cells were unusually thin in albino *Abca4*$^{-/-}$ mice (Fig. 1E).

RPE cells regularly contained undegraded phagosomes in the basal area of the cytoplasm in all three Stargardt mouse strains (Table 1). RPE cells, whose cytoplasm was largely occupied by extensive undegraded phagosomes, were found in albino *Abca4*$^{-/-}$ and *Abca4*$^{-/-}$.*Rdh8*$^{-/-}$ mice (Fig. 6B). The highest numbers of undegraded phagosomes were found in *Abca4*$^{-/-}$.*Rdh8*$^{-/-}$ mice (Table 1).

Irregularly shaped basal infoldings were frequently seen in albino *Abca4*$^{-/-}$ mice, but rarely in pigmented Stargardt mice (Table 1). The irregular shape was due to the basal infoldings expanding to the lateral side of the cells (Fig. 6C). In general, basal infoldings were reduced in number and expansion in all Stargardt mouse strains compared to wild types (Fig. 6D).

Bruch's membrane became thicker with age in all Stargardt mouse strains (Fig. 7). There was no significant difference between the different strains.

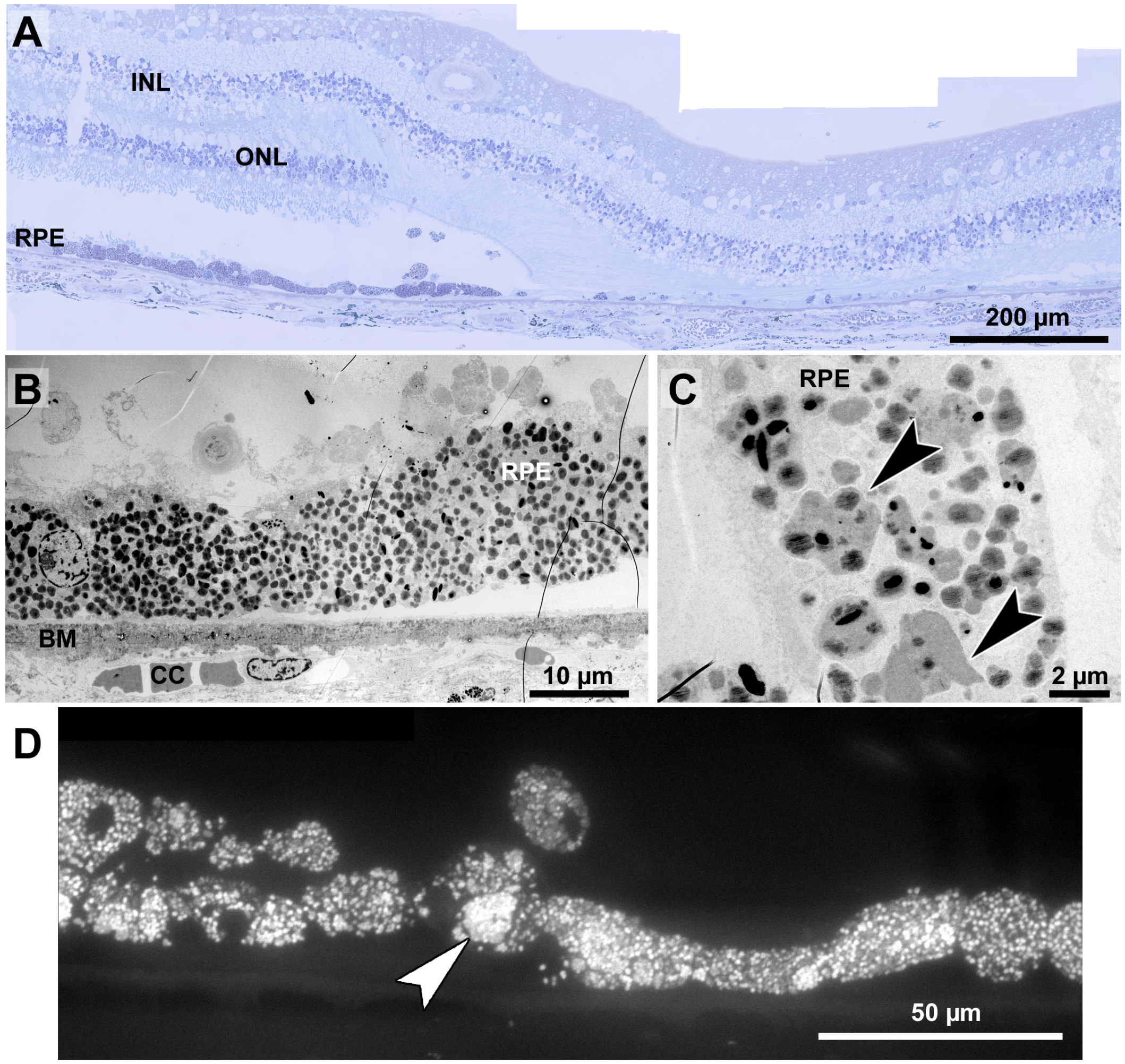

**Figure 8** **Histopathology and pigment changes in a human donor eye with late Stargardt disease.** (A) Light microscopic panorama picture of a 72-year-old human donor Stargardt retina: The left area is histologically complete with photoreceptors and a confluent RPE layer. On the right, a typical geographic atrophy with lack of photoreceptors and RPE is present. (B) Electron microscopic overview of the RPE of a human donor eye with Stargardt disease. RPE is completely filled with pigment granules. Microvilli and basal infoldings are missing and Bruch's membrane is unorganized and contains electron dense deposits. (C) Big clusters of melanolipofuscin (arrowheads). (D) Fluorescence micrograph of a semi-thin section also shows abundant lipofuscin autofluorescence and cluster formation (arrowhead). INL, inner nuclear layer; ONL, outer nuclear layer; RPE, retinal pigment epithelium; BM, Bruch's membrane; CC, choriocapillaris.

## Histopathology and pigment changes in late Stargardt disease

For comparison, we also investigated a single eye of a 72-year-old Stargardt patient (Fig. 8). In this late stage of the disease, photoreceptors were almost completely degenerated resulting in geographic atrophy as is typical for Stargardt patients (Fig. 8A). A reliable quantification of neuroretinal changes was not possible due to the long death-to-fixation time of the donor, but pigments can still be investigated under these conditions. RPE cells were nearly completely filled with pigment granules (Fig. 8B), mostly lipofuscin and

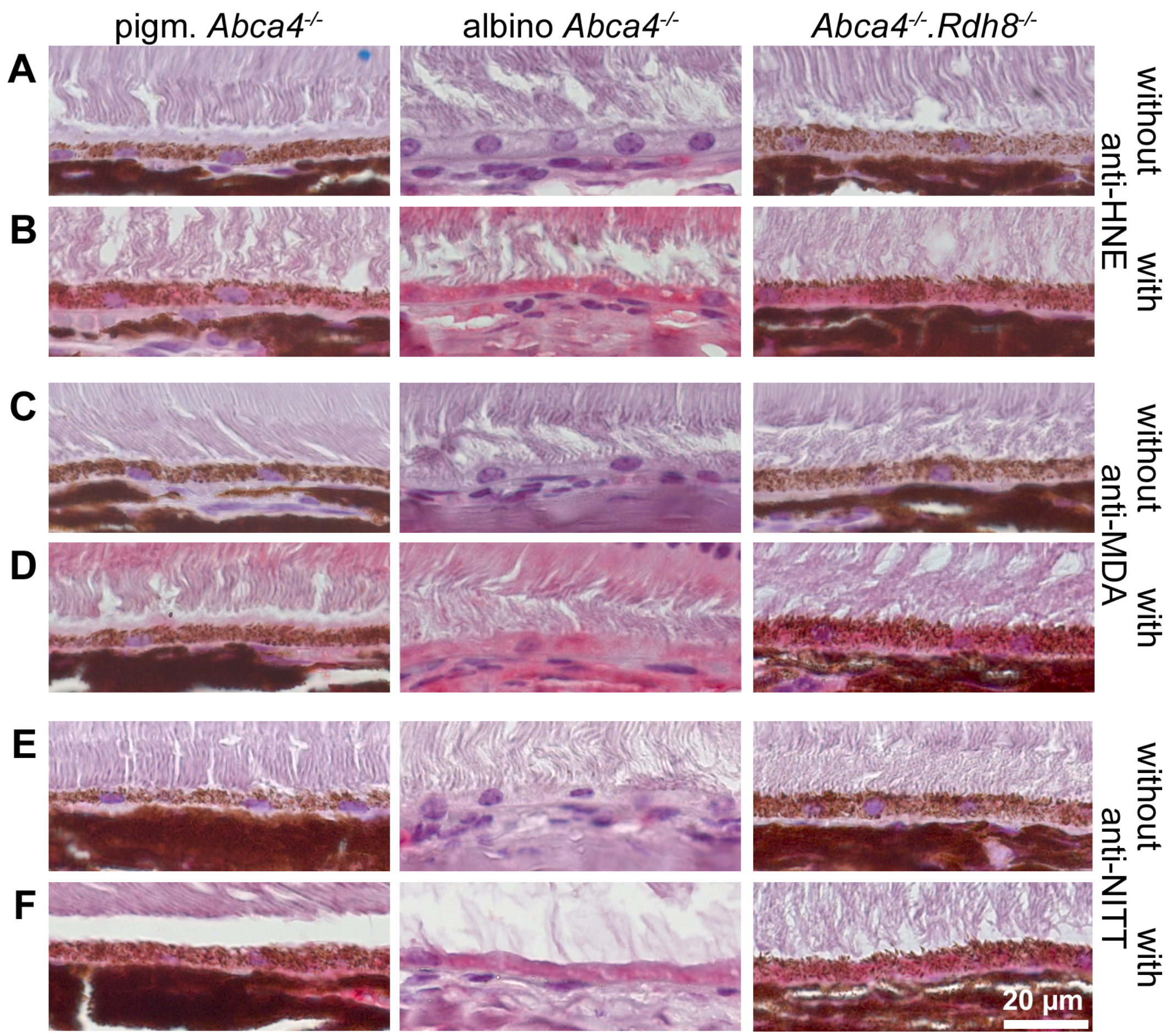

**Figure 9 HNE, MDA and NITT adducts are present in RPE cells of 12-month-old Stargardt mice indicating oxidative stress.** Immunohistochemical stainings show that (A, B) HNE, (C, D) MDA and (E, F) NITT adducts (*red*) can be found in the RPE of pigmented *Abca4*$^{-/-}$, albino *Abca4*$^{-/-}$, and *Abca4*$^{-/-}$.*Rdh8*$^{-/-}$ mice with levels being lowest in pigmented *Abca4*$^{-/-}$ mice. (A, C) and (E) show negative controls (primary antibody was omitted) for each antibody; (B, D) and (F) show the respective antibody stain for each primary antibody.

melanolipofuscin, melanin granules were barely present. Microvilli and basal infoldings were absent, while Bruch's membrane was unorganized and contained electron dense deposits (Fig. 8B). Melanolipofuscin formed clusters (Figs. 8C, and 8D), as they were also found in pigmented *Abca4*$^{-/-}$ and *Abca4*$^{-/-}$.*Rdh8*$^{-/-}$ mice.

## Oxidative stress markers are present in Stargardt mice

Since ultrastructural changes in the RPE of Stargardt mice were indicative of oxidative stress, we tested for oxidative stress markers in 12-month-old mice by immunohistochemistry. We found HNE, MDA and NITT adducts in the RPE cells of pigmented and albino *Abca4*$^{-/-}$ and *Abca4*$^{-/-}$.*Rdh8*$^{-/-}$ mice with signals being generally lowest in pigmented *Abca4*$^{-/-}$ mice (Fig. 9). HNE, MDA and NITT signals were also present in the neuroretina, reaching from the inner limiting membrane to the inner segments, in all Stargardt strains and were

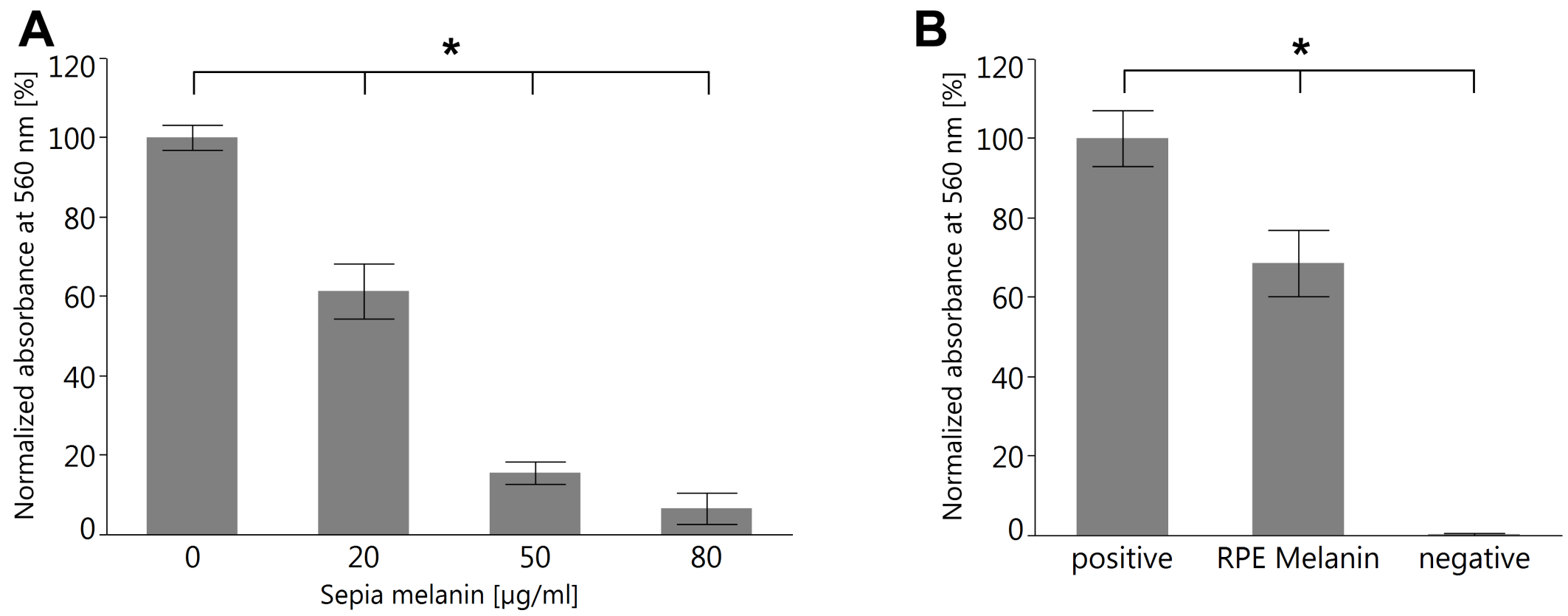

**Figure 10 Both melanin from *S. officinalis* and porcine RPE melanin can quench superoxide radicals in an NBT assay.** (A) Sepia melanin shows a dose dependent capability to quench superoxide anions produced by light-illuminated riboflavine. ($n = 8$ per group, two independent experiments, $^*p < 0.05$, Steel-Dwass All Pairs test). (B) Porcine RPE melanin (concentration $3 \times 10^4$ granules/ml) quenches superoxide anions. Negative control contained all reagents but no melanin and was kept in the dark so no superoxide was produced ($n = 12$ per group, three independent experiments, $^*p < 0.05$, Steel-Dwass All Pairs test).

highest in *Abca4*$^{-/-}$.*Rdh8*$^{-/-}$ mice in the case of HNE and MDA or at similar levels in the three strains in case of NITT (Fig. S3).

## RPE melanin can quench superoxide

To investigate the capability of RPE melanin to quench superoxide, we employed the colorimetric NBT assay (*Cheng et al., 2015*). We first tested melanin from *S. officinalis* as a melanin standard for its quenching capabilities and found a dose–response relationship (Fig. 10A). Subsequently, we investigated RPE melanin isolated from pig eyes and found an approximately 30% reduction of detectable superoxide for $3 \times 10^4$ RPE melanin granules utilized in the assay (Fig. 10B).

# DISCUSSION

The goal of this study was a comparison of the age-dependent ultrastructural changes in three frequently used mouse models for Stargardt disease in order to better understand these models. The three strains differ in their genetic background: pigmented *Abca4*$^{-/-}$ mice are bred on a 129 background, albino *Abca4*$^{-/-}$ mice have a BALB/c background and pigmented *Abca4*$^{-/-}$.*Rdh8*$^{-/-}$ mice have a mixed 129.B6 background.

Both the 129 and BALB/c strains have the Leu450 variant of RPE65, while the B6 strain has the Met450 variant, resulting in a lower visual cycle turnover and reduced light damage susceptibility in this strain. Since the pigmented *Abca4*$^{-/-}$.*Rdh8*$^{-/-}$ strain is on a mixed 129.B6 background, both alleles of RPE65 are present in this strain. Retinal degeneration was reported to be more common in *Abca4*$^{-/-}$.*Rdh8*$^{-/-}$ mice carrying the Leu450 variant (97.5% of investigated eyes) than animals carrying the Met450 variant (43.8% of investigated eyes) (*Maeda et al., 2008*). We did not genotype our animals for RPE65.

There is a growing body of evidence that different WT mouse strains show varied susceptibility to light-induced damage that is brought on by so far unidentified genetic modulators. For instance, even though BALB/c and 129 mice both have the LEU450 variant of RPE65, resulting in a high visual cycle turnover, BALB/c mice show a lower resistance to light-induced retinal damage that is independent of pigmentation (*Danciger et al., 2004*). Instead, two significant and two suggestive quantitative trait loci were identified that might be the reason for the greater sensitivity to light damage in BALB/c mice (*Danciger et al., 2004*). These findings were gained from experiments using intense, short-term light exposure, but their relevance for moderate, long-term light exposure as employed in animal husbandry is currently unknown. Nevertheless, care must be taken when comparing the results from the three different Stargardt mouse models concerning their predisposition to retinal degeneration.

## Among Stargardt mice, albino *Abca4*$^{-/-}$ mice show the earliest onset and most varied retinal damages

It was previously described that albino *Abca4*$^{-/-}$ mice show a mild retinal degeneration that is not present in albino WT mice (*Sparrow et al., 2013*; *Wu, Nagasaki & Sparrow, 2010*). We found that absolute photoreceptor nuclei numbers decrease with age, resulting in a steeper decline of photoreceptor nuclei counts compared to pigmented *Abca4*$^{-/-}$ mice (Fig. 1C). By contrast, *Abca4*$^{-/-}$.*Rdh8*$^{-/-}$ mice are described to have regional retinal degeneration and rosette formation as early as 6 weeks of age and advanced retinal degeneration by 3 months (*Maeda et al., 2008*). We were unable to reproduce the published results with our colony of *Abca4*$^{-/-}$.*Rdh8*$^{-/-}$ mice. Our animals had a similar rate of retinal degeneration than the albino *Abca4*$^{-/-}$ mice (Fig. 1C) and rosette formation was only found in one 12-month-old animal from a total of 25 *Abca4*$^{-/-}$.*Rdh8*$^{-/-}$ animals examined light-microscopically up to an age of 25 months.

In addition to the photoreceptor loss, albino *Abca4*$^{-/-}$ mice show outer segment shortening (Fig. 1B), disordered disk membranes (Fig. 2A), damaged inner segments (Fig. 2B), and morphological changes at the base of the outer segments (Figs. 2C and 3D). These damages were not seen in pigmented *Abca4*$^{-/-}$ and to a much lesser extent in *Abca4*$^{-/-}$.*Rdh8*$^{-/-}$ mice (Table 1). Shortening and disorganization of outer and inner segments was also described after conditional knockout of superoxide dismutase in the RPE, resulting in oxidative stress (*Mao et al., 2014*).

In Stargardt patients, the junction between the inner and outer segments was described to be unorganized and/or lost on SD-OCT (*Gomes et al., 2009*). Furthermore, photoreceptor anomalies were found in patients that did not display equivalent fundus autofluorescence anomalies (*Gomes et al., 2009*). Damaged inner segments and defective outer segment synthesis in albino *Abca4*$^{-/-}$ mice (Figs. 2B–2D), but not in pigmented Stargardt mouse strains, might be histological representations of the SD-OCT findings in Stargardt patients and needs further investigation.

Outer segment renewal starts at the base of the outer segment, so the disks at the tips are the oldest disks in a photoreceptor and are affected the longest by any disturbance, e.g., accumulation of retinoids due to lack of ABCA4. Loss of fundamental outer segment

proteins like rhodopsin (*Lem et al., 1999*) or GARP (*Huttl et al., 2005*) leads to a severe outer segment damage starting early in life (reviewed by (*Goldberg, Moritz & Williams, 2016*)). ABCA4 is located in the rims of disk membranes (*Papermaster et al., 1978*) and does not form stable interactions with the peripherin-2·GARP complex (*Poetsch, Molday & Molday, 2001*), which is essential for linking disk rims to the outer segment plasma membrane and disk-disk stacking (*Kaupp & Seifert, 2002*). These findings, together with the lack of outer segment disk pathology in pigmented *Abca4*$^{-/-}$ mice (Table 1) (*Charbel Issa et al., 2013*), indicate that ABCA4 is not needed for proper disk formation, orientation and stability. Whether RPE malfunction is responsible for the presented outer segment defects in albino *Abca4*$^{-/-}$ and to a lesser extent *Abca4*$^{-/-}$.*Rdh8*$^{-/-}$ mice needs to be elucidated.

## Indications for lysosomal dysfunction and oxidative stress in albino *Abca4*$^{-/-}$ mice

Formation of vacuole-like structures in the RPE as seen in albino *Abca4*$^{-/-}$ mice (Fig. 3) has been described after lysosomal integrity has been compromised and after oxidative stress. Increasing the lysosomal pH with chloroquine (*Peters et al., 2006*), inhibiting lysosomal cysteine proteases (*Okubo et al., 2000*) and a lack of functional βA3/A1-crystallin (*Valapala et al., 2014*; *Zigler Jr et al., 2011*) all lead to formation of vacuole-like structures within the RPE cell layer. Lysosomal pH was shown to be increased after treatment of RPE cells with the lipofuscin compound A2E *in vitro* (*Holz et al., 1999*; *Liu et al., 2008*) and in primary RPE cells isolated from pigmented *Abca4*$^{-/-}$ mice (*Liu et al., 2008*). Since the melanosome is a specialized lysosome (*Orlow, 1995*) that contains many lysosomal enzymes (*Diment et al., 1995*) and can fuse with phagosomes (*Schraermeyer & Heimann, 1999*), it is possible that albinism itself adds to the lysosomal dysfunction seen in albino *Abca4*$^{-/-}$ mice due to lack of melanosomes.

Similar pathologies were also described after oxidative stress. Changes in RPE morphology, such as vacuolization and loss of basal labyrinth, and deposit formation were seen after a lack of superoxide dismutase (*Mao et al., 2014*; *Seo et al., 2012*), knockout of Nrf2, a transcription factor that regulates genes involved in the cellular oxidative stress response (*Zhao et al., 2011*), and after chronic exposure to cigarette smoke (*Fujihara et al., 2008*). In some models, thinning of the outer nuclear layer (*Mao et al., 2014*; *Seo et al., 2012*) and shortening and disorganization of inner and outer segments (*Mao et al., 2014*) or RPE hyper- and hypopigmentation and choroidal neovascularization (*Zhao et al., 2011*) were also described. Besides the high oxygen partial pressure and the presence of light, other sources for oxidative stress in the RPE are phagocytosis of photoreceptor outer segments (*Miceli, Liles & Newsome, 1994*) and lipofuscin (*Rozanowska et al., 1995*).

Previous studies investigating oxidative stress in Stargardt mice found that albino *Abca4*$^{-/-}$ mice have higher levels of MDA and HNE adducts than age-matched albino WT and pigmented *Abca4*$^{-/-}$ mice (*Radu et al., 2011*) and that 6-month-old *Abca4*$^{-/-}$.*Rdh8*$^{-/-}$ mice have higher levels of 8-OHdG than 4-week-old *Abca4*$^{-/-}$.*Rdh8*$^{-/-}$ mice and age-matched WT mice (*Sawada et al., 2014*). We found similar results indicating that pigmented *Abca4*$^{-/-}$ mice have the lowest oxidative stress levels (Fig. 9).

Interestingly, *Abca4*$^{-/-}$.*Rdh8*$^{-/-}$ mice had higher numbers of undegraded phagocytosed outer segments than albino *Abca4*$^{-/-}$mice (Table 1), but albino *Abca4*$^{-/-}$ mice had higher numbers of vacuole-like structures (Fig. 3D). One reason might be higher oxidative stress levels in albino *Abca4*$^{-/-}$ mice. Melanin was shown to be a scavenger of superoxide anion and singlet oxygen (*Bustamante et al., 1993*; *Tada, Kohno & Niwano, 2010*) and protects human RPE cells from oxidative stress *in vitro* (*Burke et al., 2011*). Intravitreal application of paraquat, a superoxide anion generator, leads to higher superoxide levels and more pronounced lipid oxidation and retinal degeneration in albino compared to pigmented mice (*Cingolani et al., 2006*).

Additionally, changes of RPE morphology in albino animals might play a role in RPE vacuolization: embryonic RPE from albino mice was found to have an irregular morphology, compared to pigmented littermates (*Iwai-Takekoshi et al., 2016*). Furthermore, an altered distribution of the junctional protein P-cadherin was found in albino RPE and it was hypothesized that this might lead to defects in cell–cell adhesion (*Iwai-Takekoshi et al., 2016*). It is reasonable that a combination of weakened cell–cell contacts, impaired lysosomal function and elevated oxidative stress as described above might lead to the defects found in albino *Abca4*$^{-/-}$ mice.

## Lipofuscin varies considerably between Stargardt mouse strains

Morphological appearance of lipofuscin granules differs in the three different Stargardt mouse strains. It has been described that the chemical composition of lipofuscin in pigmented and albino WT mice differs in terms of the amounts of the different types of bisretinoids that can be found (*Ueda et al., 2016*). It is hypothesized that the higher intraocular light levels in albino animals lead to photooxidation and photodegradation of A2E and other bisretinoids (*Ueda et al., 2016*), which is in accordance with earlier publications that demonstrated lower A2E and higher A2E oxirane levels in albino *Abca4*$^{-/-}$ mice compared to pigmented *Abca4*$^{-/-}$ mice (*Radu et al., 2004*). In *Abca4*$^{-/-}$.*Rdh8*$^{-/-}$ mice, higher amounts of all-trans retinal dimer can be found than in *Abca4*$^{-/-}$ mice (*Maeda et al., 2008*). Our results show that in addition to the difference in the chemical composition of lipofuscin between the three strains, there is also a considerable histological difference.

Strikingly, lipofuscin levels in 12-month-old animals seem to be elevated in albino *Abca4*$^{-/-}$compared to pigmented *Abca4*$^{-/-}$mice, when investigated by lipofuscin autofluorescence (Fig. 4A), while quantification of RPE cytoplasm occupied by lipofuscin granules shows similar lipofuscin levels in 12-month-old pigmented and albino *Abca4*$^{-/-}$mice (Fig. 5). Both quenching effects of the melanin and differences in nature and amount of present fluorophores, as described above, are possible explanations for this seeming contradiction.

The Met450 variant of RPE65 was described to be associated with lower levels of the lipofuscin fluorophores A2E and iso-A2E in *Abca4*$^{-/-}$.*Rdh8*$^{-/-}$ mice (*Kim et al., 2004*). Since pigmented and albino *Abca4*$^{-/-}$ mice carry the Leu450 variant of RPE65 and we did not genotype the *Abca4*$^{-/-}$.*Rdh8*$^{-/-}$ mice for RPE65, lipofuscin levels in Figs. 4 and 5 in *Abca4*$^{-/-}$.*Rdh8*$^{-/-}$ mice might be underreported.

The bisretinoid A2E was shown to have surfactant-like properties and to be able to compromise lysosomal and cellular membranes (*Schutt et al., 2002*; *Sparrow et al., 1999*). In fact, we found that lipofuscin granules often lacked membranes (Fig. 4C). Furthermore, in very old animals (aged 23 months and older), we found abnormally electron dense RPE cytoplasm (Fig. 4D), making distinguishing between lipofuscin granules and cytoplasm very difficult. Whether the RPE cytoplasm gained electron density due to free lipofuscin components that leaked from lipofuscin granules after losing their membranes is unclear but appears likely.

## Bruch's membrane thickness does not vary between Stargardt mouse strains

For all three Stargardt mouse strains, a thickening of Bruch's membrane compared to WT was described (*Maeda et al., 2008*; *Radu et al., 2011*; *Weng et al., 1999*). Compared to each other, the three strains show comparable thickening of Bruch's membrane with age (Fig. 7). An increase in the thickness of Bruch's membrane is also typical in Stargardt patients (*Bonilha et al., 2016*).

## The significance of RPE melanin and its role as a scavenger of oxidative stress

The role of reactive oxygen species in retinal health was extensively investigated (*Becquet et al., 1994*; *Cingolani et al., 2006*; *Mao et al., 2014*; *Seo et al., 2012*; *Zhao et al., 2011*) and melanin was identified as a potential scavenger of reactive oxygen species (*Bustamante et al., 1993*; *Tada, Kohno & Niwano, 2010*). However, RPE melanin differs from other melanin species in several aspects. For instance, while melanocytes continuously synthesize new melanin, it is still under debate whether melanin turnover in the RPE occurs and if so, to what extent (*Schraermeyer, 1993*). Furthermore, RPE melanin can react differently than melanin in the melanocytes of the choroid (*Schraermeyer & Heimann, 1999*). Therefore, we analyzed isolated RPE melanin granules to corroborate that also this melanin species is capable to quench superoxide radicals and in fact can play a role in protection against oxidative stress in the RPE (Fig. 10). Anti-oxidative properties of RPE melanin have also been confirmed in cell culture experiments (*Burke et al., 2011*).

Certain photoreceptor and RPE-damages typical for oxidative stress and/or lysosomal impairment were exclusively found in albino *Abca4*$^{-/-}$ mice (Figs. 1D, 2B–2D, 3, 6C), while others were found in albino *Abca4*$^{-/-}$ and *Abca4*$^{-/-}$.*Rdh8*$^{-/-}$ mice, but onset was earlier in albino *Abca4*$^{-/-}$ mice (Figs. 1A–1B, 2A, 6A & 6D). Notably, even though *Abca4*$^{-/-}$.*Rdh8*$^{-/-}$ mice have a more severe disease-causing genotype compared to albino *Abca4*$^{-/-}$ mice, they only show a more pathologic phenotype regarding RPE hypertrophy (Fig. 1A, Fig. S1), the number of undegraded phagosomes (Table 1) and the percentage of RPE cytoplasm occupied by lipofuscin (Fig. 5).

### *RPE melanin affects the accumulation of lipofuscin, a major hallmark of Stargardt disease*

Melanin in the RPE plays a crucial role as scavenger for excess light and reactive oxygen species (*Hu, Simon & Sarna, 2008*). Lack of melanin has been associated with higher levels

of lipofuscin in primary RPE cells *in vitro* (*Sundelin, Nilsson & Brunk, 2001*) and a decrease of A2E with simultaneous increase of A2E oxiranes in albino compared to pigmented *Abca4*$^{-/-}$ mice (*Radu et al., 2004*). We show that albino *Abca4*$^{-/-}$ mice have higher levels of lipofuscin than pigmented *Abca4*$^{-/-}$ mice (Fig. 5).

We identified different morphologies of lipofuscin in the three different Stargardt mouse strains. In young age, pigmented *Abca4*$^{-/-}$ and *Abca4*$^{-/-}$.*Rdh8*$^{-/-}$ mice show exclusively lipofuscin with homogenous electron density (Table 1). After 12 months of age, *Abca4*$^{-/-}$.*Rdh8*$^{-/-}$ mice also have lipofuscin with flocculent electron density. Contrary, albino *Abca4*$^{-/-}$ mice have little lipofuscin with homogenous electron density, while lipofuscin with flocculent electron density is present even in young animals (Table 1).

### Aged melanin becomes pro-oxidative and might be involved in Stargardt pathology

Melanin granules probably play an essential role in the detoxification of lipofuscin due to their anti-oxidative properties (*Burke et al., 2011*). With age, melanin granules switch from expressing anti-oxidative to pro-oxidative properties after illumination (*Biesemeier et al., 2008*; *Dontsov, Glickman & Ostrovsky, 1999*; *Rozanowski et al., 2008*). This might be due to degradative modifications of melanin that occur after long-term interaction of iron-containing melanin with lipid hydroperoxides, as it happens in melanolipofuscin (*Zadlo et al., 2017*). Nevertheless, formation of melanolipofuscin might be a protective process, since radicals formed by degradation of bisretinoids (*Wu et al., 2011*) can be absorbed by the melanin moiety of melanolipofuscin and thereby protect the lysosomal membrane from damage.

In fact, alterations in near-infrared-autofluorescence, which is a marker for melanin, were identified as the earliest detectable retinal change in patients with *ABCA4*-related retinal dystrophies and can be used for predicting the further disease progress (*Cideciyan et al., 2015*). Changes in near-infrared-autofluorescence were also found in pigmented *Abca4*$^{-/-}$ mice compared to wild type (*Charbel Issa et al., 2013*). Since *Abca4*$^{-/-}$.*Rdh8*$^{-/-}$ mice have higher loads of the pro-oxidative substances lipofuscin (*Rozanowska et al., 1995*) and all-*trans*-retinal (*Zhu et al., 2016*) than pigmented *Abca4*$^{-/-}$ mice, their melanin granules might age more rapidly and switch to the pro-oxidative state earlier. This might explain the presence of lipofuscin with flocculent electron density in aged *Abca4*$^{-/-}$.*Rdh8*$^{-/-}$ mice.

### RPE melanin and its link to outer segment phagocytosis and ingestion of bisretinoids

Phagocytosis of outer segments was shown to be an oxidative stressor for the RPE (*Miceli, Liles & Newsome, 1994*). Since undegradable material from phagocytosed outer segments is transported to melanin granules in primary RPE cells (*Schraermeyer et al., 1999*; *Schraermeyer & Stieve, 1994*) and melanosomes can bind and store many toxic components (*Mecklenburg & Schraermeyer, 2007*), melanin granules might play a role in lowering phagocytosis-related oxidative or toxic stress. When post-confluent differentiated ARPE-19 cells are treated with low micromolar amounts of A2E for several weeks and then challenged with outer segments, melanin synthesis is induced (*Poliakov et al., 2014*). Poliakov et al. even proposed that A2E-induced alkalization of lysosomes could have a

physiological role in maintaining melanin pigmentation in RPE cells, since neutral pH favors eumelanogenesis (*Fuller, Spaulding & Smith, 2001*; *Ito et al., 2013*).

### *There is still much to be learned*

Taken together, all these observations point to RPE melanin being an essential factor for retinal function. Complete lack of RPE melanin granules in albino *Abca4*$^{-/-}$ mice leads to extensive retinal damage that cannot be found in pigmented tissue, even in *Abca4*$^{-/-}$.*Rdh8*$^{-/-}$ mice that have an increased amount of toxic visual cycle byproducts like all-trans retinal dimer and higher levels of lipofuscin due to their genetic burden. We cannot rule out that the lack of melanin in the iris and the choroid is an additional reinforcing factor in albino animals, but there is overwhelming data supporting the essential role of RPE melanin. Furthermore, yet unidentified differences in the genetic background of the different mouse strains might skew the presented data.

Ocular pigmentation was shown to be correlated to the incidence of uveal melanoma (*Weis et al., 2006*) and age-related macular degeneration (*Chakravarthy et al., 2010*) and tyrosinase, the key enzyme for melanin biogenesis, was found to be a potential factor for developing early stages of age-related macular degeneration (*Holliday et al., 2013*).

## CONCLUSION

In this work, we present a detailed ultrastructural comparison of the disease development of three frequently used Stargardt mouse models. As currently no animal model exists that reflects all aspects of the human disease, this will help researchers in the process of identifying which model is best suited for their research.

We found that albino *Abca4*$^{-/-}$ and *Abca4*$^{-/-}$.*Rdh8*$^{-/-}$ mice show typical histopathological signs of oxidative stress and/or lysosomal dysfunction, the earliest onset and severest changes being present in albino *Abca4*$^{-/-}$ mice. Since melanin in general is known to have anti-oxidative properties and age-dependent turnover of melanin is known to diminish the anti-oxidative properties of the RPE, we hypothesize that RPE melanin plays a crucial role in preventing Stargardt-related changes. The lack of pathology in pigmented *Abca4*$^{-/-}$ mice and the finding that also RPE melanin can quench superoxide anions support this hypothesis.

The protective role of melanin due to its antioxidative properties might be an explanation for the relatively late-onset of Stargardt disease with mild to moderate disease causing mutations (*Van Huet et al., 2014*): as long as melanin is capable of alleviating toxic effects of lipofuscin accumulation, RPE health is maintained. Once melanin is aged and lost its protective function, deleterious effects of lipofuscin accumulation can take over. This is strongly supported by the fact that changes in melanin-based near-infrared-autofluorescence temporally and spatially precede the progression of retinal degeneration in Stargardt disease (*Cideciyan et al., 2015*). Since RPE damage is a fundamental feature of Stargardt disease and age-related macular degeneration, RPE melanin should be strongly considered as a key factor in retinal health.

## ACKNOWLEDGEMENTS

The authors thank Peter Charbel Issa, Gabriel H. Travis, Roxana Radu and Krzysztof Palczewski for providing them with the Stargardt mouse strains. Special thanks are due to Foundation Fighting Blindness, Vera Bonilha and Joe Hollyfield for collecting and providing the human Stargardt tissue. In this work, the segmentation calculations were performed using Fiji on the de.NBI Cloud Tübingen (https://denbi.uni-tuebingen.de). The authors thank Jens Krüger, Maximilian Hanussek and Felix Bartusch of the High Performance and Cloud Computing Group at the Zentrum für Datenverarbeitung of the University of Tuebingen. The authors also thank Monika Rittgarn and Sigrid Schultheiss for technical assistance and Daniel Dehncke for helpful discussions.

### Funding

This work was supported by Bundesministerium für Bildung und Forschung (grant numbers 01GQ1422B and 031A535A). There was no additional external funding received for this study. The funders had no role in study design, data collection and analysis, decision to publish, or preparation of the manuscript.

### Grant Disclosures

The following grant information was disclosed by the authors:
Bundesministerium für Bildung und Forschung: 01GQ1422B, 031A535A.

### Competing Interests

The authors declare there are no competing interests.

### Author Contributions

- Tatjana Taubitz conceived and designed the experiments, performed the experiments, analyzed the data, prepared figures and/or tables, authored or reviewed drafts of the paper, approved the final draft.
- Alexander V. Tschulakow, Marina Tikhonovich, Antje Biesemeier and Sylvie Julien-Schraermeyer analyzed the data, authored or reviewed drafts of the paper, approved the final draft.
- Barbara Illing and Yuan Fang performed the experiments, authored or reviewed drafts of the paper, approved the final draft.
- Ulrich Schraermeyer conceived and designed the experiments, authored or reviewed drafts of the paper, approved the final draft.

### Human Ethics

The following information was supplied relating to ethical approvals (i.e., approving body and any reference numbers):

Ethik-Kommission an der Medizinischen Fakultät der Eberhard-Karls-Universität und am Universitätsklinikum Tübingen granted Ethical approval (approval number 462/2009BO2).

## Animal Ethics

The following information was supplied relating to ethical approvals (i.e., approving body and any reference numbers):

Einrichtung für Tierschutz, Tierärztlichen Dienst und Labortierkunde der Eberhard Karls Universität Tübingen (local agency for animal welfare) and the Regierungspräsidium Tübingen (local authorities) approved the research and all protocols involving animal handling and euthanasia.

## Data Availability

The raw data are provided in the Supplemental File.

## Supplemental Information

Supplemental information for this article can be found online at http://dx.doi.org/10.7717/peerj.5215#supplemental-information.

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
