# Peer review of "Ultrastructural alterations in the retinal pigment epithelium and photoreceptors of a Stargardt patient and three Stargardt mouse models: indication for the central role of RPE melanin in oxidative stress"

_PeerJ, doi:10.7717/peerj.5215_

## Round 0.1 · original submission · Major Revisions

This manuscript, as stated by the reviewers, is of interest to the field, but the evidence presented (see reviews) is not adequate; therefore, if you consider submitting a revised version, you should properly address the issues and concerns set forth by both reviewers, and additional evidence and detail regarding the results should be presented. Both reviewers agree that the data presented is not enough; particularly, the evidence concerning melanin and oxidative stress should be strengthened. There is already some data in the literature regarding this last point, that should also be addressed.

Reviewer 1 ·

Basic reporting

This manuscript reports on a detailed work examining retinal structures in transgenic animals. The article is very extensive, with sufficient background and literature. It is written in a style that is easy to read and understand. The paper follows the structure of the Journal.

Experimental design

The article is within the remit of the journal.

The work examines the finer details of morphological changes associated with Stargardt disease animal models. The goal is to plug the whole in our knowledge base in regards to ultra-structural alteration induced by genetic modification.

The investigators examined many aspects of the changes that is summarized in Table 1. This investigation relays on the accuracy of methodology and I have some concern about the way the tissue was prepared for examination. For the most accurate results, animals usually perfusion fixed, that ensures the best preservation of morphology. Removing the eye and post-fixed often introduces postmortem artifacts. I see no evidence that this had been controlled for. This is even more significant for the human eye samples that was fixed 48 hours death. Relaying on that tissue for fine details is unlikely to reflect real life situation, though probably the best one can lay a hand on.

It is not entirely clear whether the method used for fixation is really cap[able of preserving lipids in the tissue, and as such the lipid section appears to be over interpreted, but this might just be my lack of knowledge of the methodologies.

I am unable to judge appropriately whether the statistics are appropriate for the study, but it would be important to check this as the used method of analysis is somewhat outside the usual methods that most people use. If this is appropriate, I suggest a short explanation why this method and what the advantage over the more regular methods are.

It will be important that the Authors carefully check the manuscript and fix the inconsistencies in reporting numbers of samples used. It is very confusing for this reviewer.

Validity of the findings

This is a largely descriptive work. This is NOT a weakness, as we are lacking details of many diseases still introduce high risk methods in trying to treat the disease. We do need proper examination of diseases associated processes, and while this paper is following one line of interest in proving the detrimental effect of lipofuscin accumulation in the RPE, an idea that is not shared equally by the eye community, still add to our knowledgeable.

Whether the data is robust enough for publication, rely on how good the methodology is, and, as above, I have issues with it, but this could be only my lack of knowledge.

Whether the work on isolated pigment really adds to the work I am not entirely sure.

All in all, the question asked is inte3resting, whether there is enough evidence to clearly couple oxidative damage to the processes is somewhat less convincing without identification of molecular changes representing oxidative changes.

Additional comments

It is commendable to try to tackle issues described in this manuscript using largely morphology. The authors has a long track record in publishing high quality morphological works. There needs to be assurances that this work lives up to the standards of previous works.

Reviewer 2 ·

Basic reporting

no comment

Experimental design

no comment

Validity of the findings

They speculate that melanin protect from oxidative stress, but do not present results to support that.

Additional comments

The manuscript fit to a very interesting field related to retinal pigment epithelium (RPE) degeneration and it is clearly written. I commend the authors for their extensive data set.

The main goal of the manuscript is the evaluation among the three models; however, the data (although probably with more details) confirm previous results but are not new. In addition, in the case of the human patient, the authors showed alterations in the RPE but not that in the retina photoreceptors cells, as indicated in the title.

In figure 7, the representative electronic microscopic images of Bruch’s membrane, indicate a significant difference in the thickness of Bruch’s membrane, that have been previously described; but that is not observed in the bottom part. Then, the images are not representative.

Figure 8, can be very important. The image of the retina is not presented, although in the text indicated that photoreceptor cells are almost completely degenerated. The figure is focus on the elevated content of pigment granules and lipofuscin; but the cell structure is lost. It is indicated that Bruch´s membrane is thickened, but the dimensions should be presented and compared with a normal individual.
Discussion is extensive. I found obvious that the Abca4-/-.Rdh8-/- mice show alterations not observed in the Abca4-/-, but it is interesting that albino mice showed greater alterations than the pigmented Abca4-/-; although the speculation that melanin can protect in some way for degeneration is reasonable, the authors do not present evidence for that.
Based in the literature the authors speculated that the effects observed in RPE morphology in mutant mice are due to an oxidative stress, but they not present evidence for that. Also, a variety of studies has demonstrated melanin as a potential scavenger of reactive oxygen species, as mentioned by the authors; therefore, results on figure 9 do not support additional information.

---

## Round 0.2 · Major Revisions

Please heed the reviewer's 2 comments when preparing an amended version of the manuscript.

Reviewer 1 ·

Basic reporting

The Authors address all the issues I raised and I am satisfied with their responses.

Experimental design

Adequate to answer the questions raised.

Validity of the findings

Modifications made to clarify previous issues.

Additional comments

Congratulation for an interesting study.

Reviewer 2 ·

Basic reporting

no comment

Experimental design

no comment

Validity of the findings

no comment

Additional comments

The authors answer most of the comments, figures were improved and now they showed evidence of oxidative stress in the retina of the mutant mice.
However, I do still some comments:
1. From the results, it is clear the high variability of individuals respect to different parameters such as the thickening of Bruch membrane. I can see that the age is important; in fig.1B the 7 months albino WT, showed the same outer segments length than the Abca -/- mutant at the same age.
In this respect, in the fig. S 3, the number of photoreceptor nuclei appear similar to the WT, although disorganized ; which is the age of the mice in this figure?
2. In the human experiments, time of diagnoses. How the authors interpreted the high content of pigment granules in the RPE patient (fig 8B).
3. It is clear that melanin has an important role in RPE, as has been reported for other authors and mentioned by the authors. I think discussion is long, and now another paragraph was added, which in fact support my comment.
4. The results and discussion support the fact that other factors participate in the disease , then I consider important that based on their results the authors propose which mutant may be a better model to study the Stargardt disease.

---

## Round 0.3 · accepted · Accept

Your article has now been accepted for publication in PeerJ.

# Reviewer 2 ·

Basic reporting

No comments

Experimental design

No comments

Validity of the findings

The authors did adequate modifications

Additional comments

The authors answer all my comments ,and these were clarified
along the manuscript. I appreciate the effort of the authors.